# DENSE-RAG: Measuring and Improving Context Understanding for Consistent Retrieval-Augmented Generation

## Abstract

Retrieval-Augmented Generation (RAG) has significantly advanced LLM performance in knowledge-intensive tasks. However, when LLMs misinterpret retrieved content, they often revert to pre-trained parametric knowledge or generate hallucinated responses, undermining RAG effectiveness. In this work we try to explore this problem by proposing DEgree-based uNcertainty with Semantically Equivalent contexts (DENSE), a training-free and model-agnostic method to evaluate LLM understanding of retrieved documents. DENSE constructs semantically equivalent context and introduces a degree-based entropy to quantify response semantic uncertainty. Building on DENSE, we further introduce DENSE-RAG, which includes two training-free DENSE-guided modules: adaptive semantic chunking and iterative context refinement. Experiments on open-book QA datasets show that higher DENSE uncertainty correlates with lower QA performance, validating DENSE as a reliable indicator of LLM understanding measurement. DENSE-RAG also achieves performance competitive with state-of-the-art baselines approaches without introducing additional model or fine-tuning.

## 1 Introduction

Large Language Models (LLMs) have demonstrated remarkable success in NLP tasks. However, the reliance on parametric knowledge alone leads to knowledge cut-off issues and hallucination, making Retrieval-Augmented Generation (RAG) a crucial paradigm for knowledge-intensive tasks. In a RAG system, an information retrieval system fetches relevant documents from an external corpus, and LLMs generate the answer based on the retrieved documents. Recent research has advanced RAG by improving retrievers Shi et al. (2023); Lin et al. (2023b;a); Xu et al. (2024); Zhang et al. (2025) or through end-to-end fine-tuning of LLMs Yu et al. (2024); Izacard et al. (2023); Asai et al. (2023); Wang et al. (2024a); Huang et al. (2023). However, an essential research problem remains underexplored: how to measure LLM's understanding of the retrieved context? When LLMs fail to comprehend the input context, LLMs are observed to misinterpret contexts and make up unfaithful responses to the retrieved context Barnett et al. (2024); Song et al. (2024); Saad-Falcon et al. (2024). Consequently, the ability to assess an LLM's understanding of retrieved content represents a promising direction for evaluating and improving the reliability of RAG systems.

In this paper, we attempt to explore this problem through a semantic perspective. Consider two simple sentences, "*Bob is a physics teacher/Bob teaches physics*", which express the same meaning with different syntactic expressions. When asked "*What is Bob's occupation?*", a human reader would give the same answer regardless of which sentence is provided as context. This illustrates two key principles : (1) **the same semantics can be conveyed through different textual expressions;** and (2) **if an LLM truly understands the context, semantically equivalent inputs should yield semantically equivalent responses.**

Based on these insights, we propose **DE**gree-based u**N**certainty with **S**emantically **E**quivalent contexts (**DENSE**), for measuring LLM's understanding of contexts by evaluating the semantic consistency between multiple responses. DENSE is an **unsupervised**, **training-free** method that could be applied to any LLMs. According to the principles, we construct semantically equivalent contexts and quantify the semantic uncertainty reflected in LLM responses across these contexts. DENSE captures

semantic variation directly from response-level outputs and further enables fine-grained attribution of uncertainty to specific retrieved chunks. Experiments on open-book QA datasets show that LLMs perform significantly worse on questions with high DENSE uncertainty, demonstrating that DENSE provides a reliable indicator of contextual understanding.

On the basis of effective understanding measurement, we further leverage DENSE to enhance RAG performance. Specifically, we design two unsupervised modules: **Adaptive Semantic Chunking**, which leverages DENSE to trigger semantic chunking only under high uncertainty to improve intra-chunk semantic coherence, and **Iterative Context Refinement**, which incrementally supplements and reorganizes chunks guided by DENSE to enhance inter-chunk semantic completeness. Extensive experiments across diverse datasets and LLM backbones show that DENSE-RAG achieves competitive performance compared to state-of-the-art baselines, while offering a model-agnostic framework for both diagnosing and improving RAG systems.

Our contributions can be summarized as follows:

- We introduce **DENSE**, an unsupervised, training-free method to assess LLMs' understanding of retrieved contexts by measuring response uncertainty. Unlike prior work that primarily quantifies LLM inherent uncertainty, DENSE connect the presence of uncertainty to specific chunk, enabling targeted improvement to enhance RAG performance.

- We propose **DENSE-RAG**, which leverages DENSE to enhance RAG performance through two modules: Adaptive Semantic Chunking, which improves intra-chunk coherence under high uncertainty, and Iterative Context Refinement, which enhances inter-chunk completeness by reorganizing contexts in a DENSE-guided manner.

- We conduct extensive experiments on four open-book QA datasets with five LLMs of different scales, demonstrating that DENSE effectively evaluates LLM's understanding of contexts and is predictive of RAG performance. The proposed DENSE-RAG improves QA performance on challenging questions with high uncertainty, achieving competitive performance against state-of-the-art baselines, while maintaining flexibility and generality as a model-agnostic framework.

## 2 RELATED WORK

In Retrieval-augmented generation, a retriever Karpukhin et al. (2020); Douze et al. (2024) is employed to obtain relevant document chunks from an external corpus, then LLM takes the retrieved context to generate replies Gao et al. (2023). Enhancing LLMs' understanding of retrieved documents to improve the overall alignment of the system remains a significant challenge. Some works improve retrievers to align the needs of LLMs Shi et al. (2023); Lin et al. (2023b;a) or add-on moderate-size models Xu et al. (2024); Zhang et al. (2025). Despite providing stronger retrievers, one potential approach is to finetune LLM in an end-to-end manner Yu et al. (2024); Izacard et al. (2023); Asai et al. (2023); Wang et al. (2024a); Huang et al. (2023); Yoran et al. (2023).

Enhancing the reliability of the generation by measuring the uncertainty in LLM responses has emerged as a promising direction Kuhn et al. (2023); Farquhar et al. (2024); Hou et al. (2024); Jiang et al. (2024). Semantic uncertainty was proposed to estimate uncertainty in language generation tasks in an unsupervised manner Kuhn et al. (2023). By quantifying the semantic differences among the responses, researchers can effectively measure the impact of hallucinations in LLMs Farquhar et al. (2024). Hou et al. (2024) proposed a method to decompose uncertainty by generating clarifications and ensembling. While many works discuss the uncertainty in LLMs in unsuperivsed scope Lin et al. (2024); Jiang et al. (2024), some works also try to identify uncertainty and improve the performance of LLMs in a supervised manner Kweon et al. (2025); Liu et al. (2024a); Arteaga et al. (2025). Several works have investigated how LLM uncertainty manifests in RAG settings. Dai et al. (2025) quantify the utility of retrieval by capturing LLM's internal belief in RAG scenarios. Hasegawa et al. (2024) measured certainty in retrieval and generation seperately through Rouge-L or the BERT score. Perez-Beltrachini & Lapata (2025) trained a passage utility model to predict the utility of each passage in the context of LLMs. However, these studies often focus on how to measure uncertainty within the system, while how to effectively link uncertainty to the retrieved documents and leverage it to improve the performance of RAG systems remains largely underexplored.

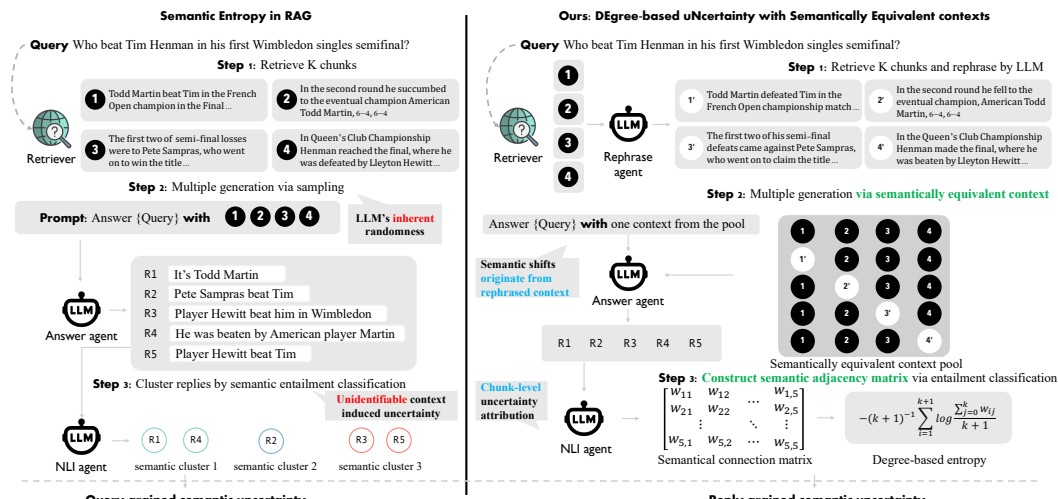

Figure 1: Overview of DENSE. DENSE leverage context rephrasing and degree-based semantic uncertainty to (1) ensure that all semantic variation in the output arises from the LLM's interpretation of the rephrased chunks, and (2) identify the specific chunks responsible for such variation.

## 3 PRELIMINARIES

To lay the groundwork for analyzing LLMs' understanding of retrieved content, we first introduce the RAG task formulation, semantic-related formulations and properties. Given a question $q$, a retriever fetches top-$k$ documents from a knowledge base to construct context $C = [c_1, c_2, ..., c_k]$ that are most relevant to $q$. LLM $f_\theta$ is used to produce the response $r$:

$$r = f_\theta(q, C), \tag{1}$$

where $r$ could be a short phrase or sentence. Kuhn et al. (2023) discuss the meanings and forms of natural language, *"Although models' input is words, but for almost all applications we care about meanings"*. This observation underlines the central role of semantics in NLP tasks, we summarize the relationship between semantic meaning and retrieved context in RAG as two formulations.

**Formulation 1**: *For natural language context $C$ and question $q$, there is a semantic space $\mathcal{S}$ and a mapping function $\pi$, that maps $q, C$ to its underlying semantic $\pi(q, C)$ in semantic space $\mathcal{S}$.*

**Formulation 2**: *If there exists $C' \neq C$ such that $\pi(q, C') = \pi(q, C)$, then $C'$ and $C$ are semantically equivalent under question $q$ in the semantic space $\mathcal{S}$.*

According to these formulations, two different textual contexts $C$ and $C'$ can yield the same semantics for a given question $q$. If a human reader or an LLM fully understands $C$, it should also produce a semantically equivalent response when given $C'$. This property is commonly referred as semantic consistency in prior work Rabinovich et al. (2023). Accordingly, we extend this notion of semantic consistency in the RAG setting and propose the following property:

**Property 1**: *If an LLM $f_\theta$ can understand $C$ while answering question $q$. With $\pi(q, C) = \pi(q, C')$, the LLM's responses under $C$ and $C'$ should be semantically equivalent:*

$$\pi(q, C) = \pi(q, C') \rightarrow \pi(f_\theta(q, C)) = \pi(f_\theta(q, C')) \tag{2}$$

This property highlights that if an LLM produces diverse semantic under semantically equivalent contexts, the discrepancy signals a misalignment in its interpretation of input semantics.

## 4 EVALUATING LLM'S UNDERSTANDING OF RETRIEVED CONTEXT

Building on the aforementioned property, we can examine whether the LLM has adequately understood the retrieved context by measuring its semantic variance under $C$ and $C'$. As we shown in

Figure 1, prior work on semantic uncertainty in language generation focuses on sampling-based decoding to probe LLM's inherent randomness (Kuhn et al., 2023; Lin et al., 2024) rather than uncertainty arising from the retrieved context. Moreover, these methods produce query-level uncertainty scores and cannot indicate which retrieved chunks drive semantic variance. Therefore, we introduce DENSE (DEgree-based uNcertainty with Semantically Equivalent contexts), to evaluate LLM's understanding of contexts. DENSE consists of two main steps: we first construct semantically equivalent contexts through rephrasing and obtain responses via greedy decoding to isolate the influence of LLM sampling; then degree-based uncertainty measure is proposed to capture semantic variations across responses, which enables us to localize the chunks that contribute to the uncertainty.

## 4.1 SEMANTICALLY EQUIVALENT CONTEXT REPHRASING

We first construct semantically equivalent and textually diverse contexts $C'$. Specifically, we use an LLM to rephrase each retrieved chunk in isolation so that any semantic variation in the model's responses can be attributed to a single chunk without cross-chunk interference. For a retrieved context $C_0 = [c_1, \ldots, c_k]$, we obtain a rephrased $c'_i$ for each chunk $c_i$:

$$c' = f_\theta(p_r(c)), \ where \ \pi(c') = \pi(c), \ and \ c' \neq c \tag{3}$$

$p_r$ is the rephrasing prompt (see Appendix B.1). This yields a set of rephrased chunks $\{c'_1, \ldots, c'_k\}$. We then construct $k$ single-edit contexts by replacing exactly one chunk at a time:

$$C_i[j] = \begin{cases} c'_i, & \text{if } j = i \\ c_j, & \text{otherwise} \end{cases} \quad \text{for } j = 1, \ldots, k \tag{4}$$

In this manner, we obtain $k+1$ contexts in total: the original $C_0$ and $k$ semantically equivalent variants $C_1, \ldots, C_k$. To ensure that rephrasing preserves the original semantics, we check the semantic shift between rephrased chunks and their original counterparts. The detailed results are shown in Appendix D. Over 95% of the rephrased chunks are identified as semantically entailment to the originals. This demonstrates that the LLM is fully capable of performing faithful rephrasing, and it introduces only negligible semantic variation within the DENSE framework. To ensure that rephrasing preserves the original semantics, we also empirically compare QA results with and without rephrasing in Section 6.2, demonstrating the robustness of LLM rephrasing.

By greedy decoding, we disentangle LLM's inherent randomness ensuring that when the input is identical, the LLM always produces the same output. This guarantees that any semantic variation observed in our experiments arises solely from the LLM's understanding of rephrased contexts. Supporting experiments and detailed discussion are provided in Appendix C. Under this setup, generating with semantically equivalent contexts yields a reply set $R = \{r_0, r_1, \ldots, r_{k+1}\}$.

## 4.2 DEGREE-BASED SEMANTIC UNCERTAINTY

Given reply set $R$ under semantically equivalent contexts, the next step is to assess the semantic variation in the LLM responses. To this end, we propose Degree-based Semantic Entropy, an effective method to quantify semantic uncertainty across these responses.

Semantic entropy was introduced by Kuhn et al. (2023), which measures uncertainty by clustering responses into semantic groups. However, since each generation in our setting is conditioned on a rephrased context, we need to further identify semantic variations introduced at the chunk level. In this case, semantic entropy becomes inadequate for capturing such reply-grained uncertainty.

We therefore propose degree-based semantic entropy to compute entropy directly in response level. Instead of clustering, we treat each response as a graph node, and construct a semantic adjacency matrix $W$ using the entailment scores between multiple responses Jiang et al. (2024); Lin et al. (2024):

$$w_{ij} = ((NLI(r_i, r_j) + NLI(r_j, r_i))/2)_{i,j \in [0, \ldots, k]}, \tag{5}$$

where a natural language inference model(NLI) is used to classify whether $r_i$ and $r_j$ are *entailment(1)* or *neutral(0)*, $w_{ij}$ represents the link between two responses. In DENSE, an LLM is employed to make the classification and the prompt is in Appendix B.2. After constructing $W$, we compute the degree-based semantic uncertainty as follows:

$$DSE(q, C) = -(k+1)^{-1} \sum_{i=1}^{k+1} \log \frac{D_i}{k+1}, \tag{6}$$

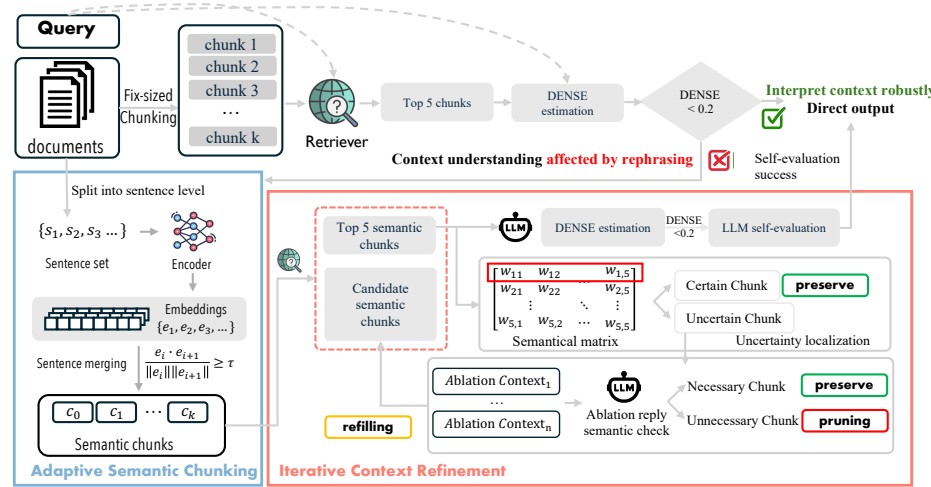

Figure 2: Overview of DENSE-RAG framework. Adaptive semantic chunking improves intra-chunk coherence, and iterative context refinement enhances inter-chunk completeness.

where $D_i = \sum_{j=0}^{k} w_{ij}$, is the degree of response $r_i$. The degree of a response represents how many other responses are semantically aligned with it in the graph, corresponding to the number of adjacent nodes in the semantic graph. $\frac{D_i}{k+1}$ represents the average link strength of response $r_i$ with respect to all other responses. Degree-based semantic entropy is a non-clustered variant of semantic entropy, we discuss the relationship of DSE and semantic entropy in Appendix E and provide the pseudocode of the DENSE in Appendix F.1.

We empirically verify that DENSE effectively reflects LLMs' understanding of retrieved contexts, where higher DENSE scores consistently correlate with worse QA performance. The experimental setup, results, and corresponding discussions are presented in Section 6.

## 5 IMPROVING CONTEXT QUALITY WITH DENSE GUIDANCE

In the previous section, we proposed DENSE as an indicator of the LLM's ability to understand retrieved contexts. Building on this foundation, we move beyond measurement and leverage DENSE to improve context quality. Since higher uncertainty indicates worse performance, we categorize questions into **certain** ($DENSE \leq 0.2$) and **uncertain** ($DENSE > 0.2$). We propose two **model-agnostic, training-free** modules that enhance intra-chunk semantic consistency and inter-chunk completeness, thereby improving LLM performance on uncertain questions.

### 5.1 ADAPTIVE SEMANTIC CHUNKING

Chunking strategies in RAG face a fundamental trade-off: fixed-size chunking is efficient but often splits contextually related sentences, disrupting semantic coherence Gao et al. (2023); Finardi et al. (2024); semantic chunking, which groups sentences by embedding similarity, can better preserve semantic but its computational cost frequently outweighs its performance gains Qu et al. (2024).

To address this trade-off, we design a DENSE-driven adaptation mechanism that selectively applies semantic chunking only when high uncertainty is detected under fixed-size chunking. The intuition is that uncertain questions are more likely to suffer from semantic inconsistencies in the retrieved chunks, and thus benefit more from semantic chunking. As illustrated in Figure 2, given a question $q$, we first run fixed-size chunking on the given documents $d$, use the top-$k$ chunks for DENSE evaluation, and classify $q$ as certain or uncertain. If $q$ is a certain question, we directly output the response. Otherwise, we split the original documents into sentences $d = \{s_1, s_2, s_3, ...\}$, encode each sentence $s_i$ into a vector $e_i$, and iteratively merge $s_{i+1}$ into chunk $c_j$ if

$$\frac{e_i \cdot e_{i+1}}{\|e_i\|\|e_{i+1}\|} \geq \tau \; and \; |c_j \cup s_{i+1}| \leq T_{max}, \tag{7}$$

where $\tau$ is the similarity threshold and $T_{max}$ is the maximum token length for a chunk. If the condition is not satisfied, $s_{i+1}$ starts a new chunk $c_{j+1}$. By adaptive semantic chunking, we change the chunking strategies while LLM have a uncertain understanding on fixed-size chunking. And during the adaptive semantic chunking, we only change the chunking strategy rather than introducing additional documents. The pseudocode of adaptive chunking is provided in Appendix F.2. We evaluate the effectiveness of Adaptive Chunking in Section 6. Our adaptive mechanism preserves the strong performance of fixed-size chunking on certain cases while selectively leveraging semantic chunking to enhance QA performance on uncertain ones.

## 5.2 ITERATIVE CONTEXT REFINEMENT

Semantic inconsistency can occur both within and across chunks. For complex questions, it is often the case that an individual retrieved chunk is insufficient to answer the query, highlighting the need for better inter-chunk coherence. To address this, we propose a DENSE based iterative context refinement module. The module evaluates retrieved chunks using DENSE and categorizes them into three types—certain, necessary, and unnecessary, based on their semantic contribution when serving as context. Then the module refine context by retaining certain chunks, removing unnecessary ones, and supplementing the context with new chunks guided by the necessary ones.

**Localize the source of uncertainty.** Since DENSE computes semantic adjacency $W$, and each modified context $C_i$ differs from the original $C_0$ in exactly one rephrased chunk $c_i'$, we can evaluate the impact of each $c_i$ and classify it as *certain* or *uncertain*:

$$l_i^{ce} = \mathbb{1}_{\{w_{i,0}=1\}}, \tag{8}$$

where $\mathbb{1}$ denotes the indicator which equals 1 if $w_{i,0} = 1$ and 0 otherwise. If $l_i^{ce} = 1$, chunk $c_i$ is classified as a *certain chunk*, indicating that even after rephrasing, the corresponding response $r_i$ remains semantically consistent with the original response $r_0$. This suggests that the LLM's understanding of $c_i$ is robust and unaffected by rephrasing. Conversely, an *uncertain chunk* indicates that the LLM fails to correctly capture the intended semantics when that chunk is rephrased.

After distinguishing between certain and uncertain chunks, we further analysis the uncertain ones. Semantic uncertainty in LLM responses under a rephrased chunk can stem from two different scenarios: (i) the chunk is topically relevant but lacks the answer, leading to uncertainty due to incomplete information, or (ii) the chunk is weakly relevant(maybe total irrelevant) and contains noise, which misleads the model and introduces spurious uncertainty. To differentiate these two scenarios, we perform an ablation generation by masking each uncertain chunk:

$$ar_i = f_\theta(q, [c_1, ..., c_{i-1}, c_{i+1}, ..., c_k]), \tag{9}$$

where $ar_i$ denotes the response when chunk $c_i$ is absent. Then we employ the entailment in Equation 5 to evaluate whether chunk $c_i$ is necessary:

$$l_i^{ne} = \mathbb{1}_{\{[NLI(r_0,ar_i)+NLI(ar_i,r0)]/2=0\}}, \tag{10}$$

where $r_0$ is the response under original context. If removing an *uncertain chunk* changes the model's answer, it implies that the chunk carries critical information, and we classify it as a *necessary chunk*. Conversely, if its removal does not affect the answer, the chunk is *unnecessary*, as it is either irrelevant or redundant. Through this process, DENSE together with ablation allows us to classify each chunk $c_i$ into three types: *certain* ($l_i^{ce} = 1$), *necessary* ($l_i^{ce} = 0, l_i^{ne} = 1$), and *unnecessary* ($l_i^{ce} = 0, l_i^{ne} = 0$).

**Iterative refinement.** After categorizing all chunks, we refine the context with two steps: (i) pruning, which removes *unnecessary chunks*, and (ii) refilling, which adds new chunks most similar to the *necessary chunks* based on cosine similarity of embeddings. The refinement proceeds iteratively and after each update, we recompute DENSE and stop once either (a) DENSE falls below 0.2, indicating LLM understands context with certainty, or (b) the LLM evaluates the context as sufficient in self-evaluation. If neither condition is met after all candidate chunks are explored, we fall back to the subset of chunks yielding the lowest DENSE score. The self-evaluation prompt is described in Appendix B.3 and the complete pseudocode is provided in Appendix F.3.

## 6 EXPERIMENTS

In this section, we conduct experiments to demonstrate the effectiveness of DENSE-RAG and analyze the contributions of each component. First we validate DENSE as an indicator of contextual under-

standing in Section 6.2. In Section 6.3, we evaluate how DENSE-RAG improves QA performance on uncertain questions across different LLM backbones as well as comparing with other baselines. Section 6.4 presents ablation studies to examine the design choices of adaptive semantic chunking and iterative context refinement. We also include additional robustness demonstration, sensitivity analyses and case studies in Appendix K and Appendix M.

## 6.1 EXPERIMENT SETUP

**Datasets.** We test our methods on open-book QA datasets, which require free-form answers: TriviaQA Joshi et al. (2017), Natutal Question Kwiatkowski et al. (2019), AmbigNQ Min et al. (2020) and 2WikiQA Ho et al. (2020). The first three are single-hop QA datasets, while 2WikiQA is a multi-hop QA dataset. We use Exact Match (EM) as the metric to evaluate QA performance. The detailed information is in Appendix G.

**Implementation Details.** We conduct experiments on Qwen-2.5 1.5B, Qwen-3 8B, Llama-3 8B, Llama-3.1 8B, and Llama-3.1 70B, using the documents provided by each dataset as the retrieval corpus. A vanilla RAG pipeline is built with recursive chunking (chunk size = 512) as the default strategy. For dense retrieval, we adopt UAE-Large-V1 as the encoder for both questions and documents, and use FAISS for indexing. Unless otherwise specified, the top-5 retrieved chunks are used as context in all experiments. We also conduct experiments with different number of chunks in Appendix K. Details of each component and the QA prompt are provided in Appendix B.4 and H.

## 6.2 DENSE AS A MEASURE OF CONTEXT UNDERSTANDING

We first demonstrate that DENSE is an effective way to quantify LLM's understanding of retrieved context. Followed prior work Kuhn et al. (2023) settings, when the LLM understands the semantically equivalent contexts, the responses tend to be *more consistent*, and are more likely to be *correct*. We compute DENSE on a Llama-3.1 8B vanilla RAG, and present the average exact match within different DENSE intervals in Figure 3. The results show that average exact match decreases as DENSE increases, confirming that higher semantic uncertainty corresponds to lower QA accuracy and that **DENSE provides an effective unsupervised measure of LLMs' contextual understanding**.

To enable a comparison between DENSE and other uncertainty estimation methods for natural language tasks, we perform evaluations under sampling with temperatures 0.25 and 0.5, and compare DENSE against five uncertainty baselines using AUROC and AURAC (Table 1). DENSE is specifically proposed to work in greedy decoding mode to isolate the influence of inherent LLM randomness, ensuring that the measured uncertainty primarily reflects variations in contextual understanding. Even under this design constraint, our method still consistently outperforms other baselines in sampling settings. This demonstrates that DENSE is better aligned with the retrieval-grounded nature of RAG, effectively capturing uncertainty rooted in how the LLM interprets retrieved evidence rather than in generation randomness.

We additionally conduct experiments at higher temperatures. As temperature increases, RAG accuracy deteriorates sharply and the LLM gradually ceases to follow the retrieved context. As a result, the semantic variation in its responses becomes dominated by internal sampling noise rather than differences in contextual comprehension. The experiment results at high temperatures and the detailed discussion are provided in the Appendix I.1.

To verify that performance drops occur across various LLM backbones, we conduct RAG experiments on Qwen-2.5 1.5B, Llama-3.1 8B and Llama-3.1 70B without DENSE, comparing their performances on certain and uncertain questions in Figure 4. The consistent performance drop confirms that DENSE provides a reliable measurement. The comparison between RAG-DENSE$_{eval}$ 8B and Llama3.1 RAG 8B in Figure 4 shows that rephrasing in DENSE has negligible impact on QA performance, which verifies that our rephrasing process does not cause semantic drift in the chunks. Additional results, including experiments on summarization datasets, are presented in Appendix I.

### 6.2.1 ROBUSTNESS OF DENSE

We evaluate the robustness of DENSE by testing multiple thresholds for separating certain/uncertain questions, as shown in Figure 5. Across all thresholds, the performance gap between the two groups remains significant, confirming that DENSE measurement is stable and effective. For our main

Table 1: AUROC and AURAC under greedy decoding and low-temperature sampling. DENSE achieves consistently higher performance than baselines designed for language generation task, indicating that DENSE is more effective in RAG settings.

| Uncertainty Measurements | TriviaQA | | Natural Question | | AmbigNQ | | 2WikiQA | |
|---|---|---|---|---|---|---|---|---|
| | AUROC↑ | AURAC↑ | AUROC↑ | AURAC↑ | AUROC↑ | AURAC↑ | AUROC↑ | AUARC↑ |
| *greedy decoding* | | | | | | | | |
| $DENSE_{llm}(ours)$ | 75.49 | 92.69 | 67.26 | 75.83 | 68.20 | 78.06 | 65.02 | 66.27 |
| $DENSE_{deberta}(ours)$ | 66.94 | 91.02 | 59.42 | 73.69 | 58.01 | 74.89 | 53.49 | 61.42 |
| *temperature=0.25* | | | | | | | | |
| Discrete Semantic Entropy Farquhar et al. (2024) | 64.07 | 88.72 | 61.33 | 72.29 | 60.40 | 75.94 | 57.28 | 60.80 |
| $U_{eigv}$ Lin et al. (2024) | 66.64 | 89.05 | 64.16 | 73.08 | 62.90 | 76.01 | 57.37 | 60.43 |
| $U_{deg}$ Lin et al. (2024) | 66.56 | 89.03 | 64.05 | 73.69 | 62.74 | 76.79 | 57.25 | 60.82 |
| KLE heat$_{t=0.1}$ Nikitin et al. (2024) | 66.40 | 89.00 | 63.93 | 73.89 | 62.53 | 75.84 | 56.99 | 60.36 |
| KLE deberta matern$_{\kappa=3.0,\nu=3.0}$ Nikitin et al. (2024) | 63.97 | 89.11 | 61.98 | 72.51 | 62.33 | 77.56 | 56.78 | 61.05 |
| $DENSE_{llm}(ours)$ | **75.63** | **92.52** | **67.96** | **75.10** | **69.94** | **79.54** | **65.74** | **66.38** |
| *temperature=0.50* | | | | | | | | |
| Discrete Semantic Entropy Farquhar et al. (2024) | 71.15 | 90.56 | 65.59 | 74.46 | 66.61 | 78.79 | 61.50 | 63.57 |
| $U_{eigv}$ | 74.32 | 91.90 | **69.35** | **77.08** | 68.79 | 77.47 | 61.97 | 63.93 |
| $U_{deg}$ | 74.24 | 91.53 | 69.28 | 76.82 | 68.64 | 77.89 | 61.84 | 63.44 |
| KLE heat$_{t=0.1}$ Nikitin et al. (2024) | 73.87 | 91.48 | 69.18 | 76.79 | 68.04 | 77.78 | 61.41 | 63.72 |
| KLE deberta matern$_{\kappa=3.0,\nu=3.0}$ Nikitin et al. (2024) | 68.84 | 90.72 | 64.57 | 75.09 | 63.25 | 77.95 | 60.17 | 63.54 |
| $DENSE_{llm}(ours)$ | **77.53** | **93.06** | 69.23 | 76.71 | **72.02** | **81.28** | **67.28** | **68.08** |

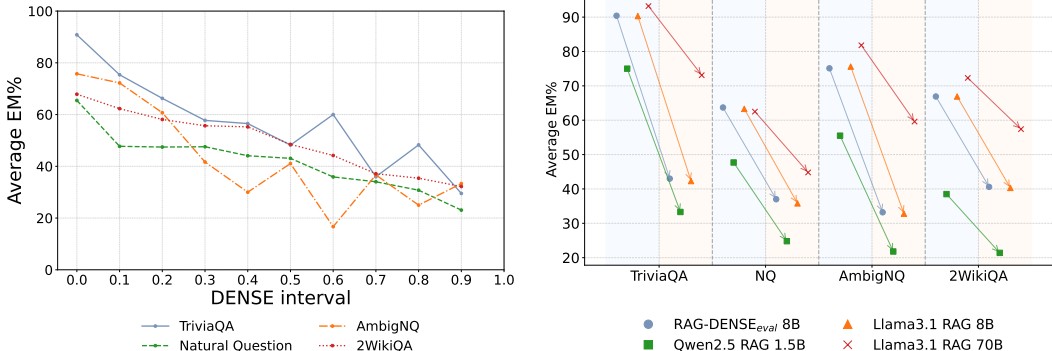

Figure 3: Average EM across different DENSE intervals. On four Open-book QA tasks, average EM decreases as DENSE increases.

Figure 4: Exace match on certain (blue: DENSE $\leq 0.2$) and uncertain (ivory: DENSE $> 0.2$) questions across LLM backbones of different scales.

experiments, we adopt 0.2 as the default threshold, supported by the observation from Figure 3 that across all datasets, the average exact match decreases monotonically as DENSE increases when $DENSE \leq 0.2$. In different applications, the choice of threshold can be flexible. In domains requiring higher certainty, such as healthcare or law, a lower threshold enforces more certain outputs but classifies more questions as uncertain, triggering chunking and refinement more frequently. Higher thresholds reduce computation at the cost of tolerating greater semantic variability.

To assess whether DENSE depends on a particular NLI backbone, we replace the LLM-based entailment judge with a supervised model, DeBERTa-large-MNLI. We evaluate the DeBERTa-based DENSE using the same AUROC and AURAC metrics. As shown in Table 1, DENSE maintains consistently strong performance across datasets under both NLI backbones, indicating that the method is not tied to LLM-specific behaviors and generalizes well when using a small supervised NLI model. This robustness suggests that DENSE captures intrinsic semantic uncertainty rather than artifacts of any particular entailment model.

### 6.3 DENSE-RAG QA PERFORMANCE

**DENSE-RAG is effective on uncertain questions**. To evaluate the effectiveness of DENSE-RAG, we progressively incorporate adaptive semantic chunking and iterative context refinement into RAG pipeline. The results on uncertain questions are summarized in Table 2. We observe consistent gains as each module is added, with the full DENSE-RAG achieving improvements across uncertain question. For Qwen-2.5 1.5B, the limited parameter size makes it inherently difficult to handle multi-hop reasoning, which is also evident when compared with other LLMs. In contrast, Llama-3.1 70B is already very strong, so the gain on AmbigNQ is marginal.

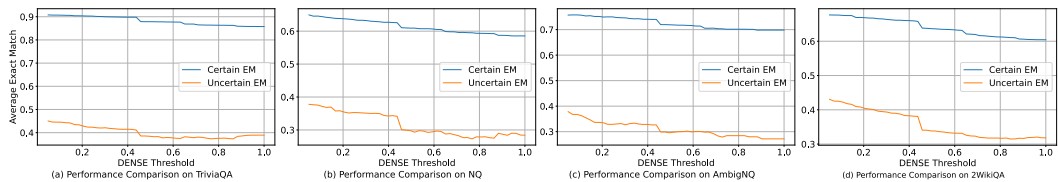

Figure 5: Average exact match on certain vs. uncertain questions under different DENSE thresholds, showing a consistent performance gap between the two groups.

Table 2: Experimental result on **uncertain questions** on 4 datasets. Each component shows improvement across LLMs of different scales. Chunking and Refinement denotes the proposed adaptive semantic chunking and iterative context refinement modules.

| Backbone | DENSE Component | | TriviaQA | Natural Question | AmbigNQ | 2WikiQA |
|---|---|---|---|---|---|---|
| | Chunking | Refinement | | | | |
| | | | 33.28 | 24.79 | 21.82 | **21.39** |
| Qwen-2.5 1.5B | ✓ | | 35.27 | 27.00 | 24.09 | 21.19 |
| | ✓ | ✓ | **36.36** | **27.08** | **24.55** | 21.33 |
| | | | 46.51 | 35.35 | 33.18 | 42.03 |
| Qwen-3 8B | ✓ | | 49.42 | 37.73 | 36.36 | 42.68 |
| | ✓ | ✓ | **49.83** | **38.25** | **37.73** | **44.35** |
| | | | 42.60 | 35.60 | 32.27 | 40.57 |
| Llama-3 8B | ✓ | - | 51.33 | 38.93 | 40.00 | 41.73 |
| | ✓ | ✓ | **53.91** | **40.54** | **41.36** | **45.78** |
| | | | 42.26 | 35.78 | 32.27 | 40.30 |
| Llama-3.1 8B | ✓ | | 52.41 | 41.31 | 40.45 | 42.34 |
| | ✓ | ✓ | **56.32** | **42.16** | **43.63** | **46.55** |
| | | | 73.12 | 44.80 | 59.55 | 57.37 |
| Llama-3.1 70B | ✓ | | 74.63 | 45.49 | **61.36** | 59.85 |
| | ✓ | ✓ | **74.87** | **46.93** | 60.00 | **61.20** |

**DENSE-RAG achieves competitive performance against SOTA RAG.** We compare DENSE-RAG with state-of-the-art baselines in Table 3. At the 8B scale, DENSE-RAG achieves performance comparable to finetuned systems such as RankRAG Yu et al. (2024). Although RAG-DDR outperforms DENSE on TriviaQA, it is an end-to-end trained framework, whereas DENSE-RAG requires no additional training and can be flexibly integrated into diverse RAG applications. Results on all five backbones and more baseline comparisons are provided in Appendix J.

DENSE-RAG evaluates an LLM's understanding of retrieved context and improves QA performance without any training or access to model internals, while keeping the overall computational cost within $O(k^2)$ LLM calls, where $k$ is the number of retrieved chunks. The detailed complexity of each module is further discussed in Appendix L.

## 6.4 ABLATION STUDY

To better understand the contributions of individual designs in DENSE-RAG, we conduct a set of ablation studies and comparative experiments. We focus on two main aspects: (i) the impact of different chunking strategies on performance under certain and uncertain questions, and (ii) the effectiveness of the iterative context refinement and its key components under varying configurations.

### 6.4.1 ANALYSIS OF CHUNKING STRATEGIES FROM UNCERTAINTY PERSPECTIVE

Chunk size introduces a natural trade-off: larger chunks preserve more information, while smaller ones reduce noise Zhang et al. (2025). Although semantic chunking has been proposed to improve coherence, prior work reports inconsistent gains compared to fixed-size chunking Qu et al. (2024). To examine this issue from an uncertainty perspective, we compare: (1) fixed-size recursive chunking with 256/32 and 128/16 settings, and (2) universal semantic chunking. Experiment results on certain and uncertain questions are shown in Table 4, which leads to the following interesting findings:

**Uncertain questions benefit from smaller chunks.** Reducing chunk size improves performance on uncertain questions but reduces accuracy on certain ones (Table 4). For certain questions, larger chunks maintain robustness by providing sufficient context despite added noise. In contrast, for uncertain questions, smaller chunks help filter irrelevant information, yielding marginal gains.

Table 3: Comparison of DENSE-RAG with baselines. Ret. FT and Gen. FT indicate whether the retriever and generator of the method were fine-tuned, respectively.

| Method | Generator Model | Ret. FT | Gen. FT | TriviaQA | Natural Question | AmbigNQ | 2WikiQA |
|---|---|---|---|---|---|---|---|
| RePlug-LSR (few-shot)Shi et al. (2023) | Codex 175B | ✓ | ✗ | 77.3 | 45.5 | - | - |
| ChatQA-1.5Liu et al. (2024b) | Llama 8B | ✓ | ✓ | 81.0 | 42.4 | - | 26.8 |
| UncertaintyRAGLi et al. (2024b) | Llama2 13B | ✓ | ✗ | 82.5 | - | - | 38.3 |
| ERM4Shi et al. (2024) | GPT-3.5-turbo | ✓ | ✗ | - | 52.7 | 53.5 | 46.8 |
| Astute-RAGWang et al. (2024b) | Claude 3.5 Sonnet | ✗ | ✗ | 84.5 | 53.6 | - | - |
| RankRAGYu et al. (2024) | Llama3 70B | ✗ | ✓ | 85.6 | 54.2 | - | 38.2 |
| RAG-DDRLi et al. (2024a) | Llama3 8B | ✓ | ✓ | 89.6 | 52.1 | - | - |
| DENSE-RAG | Llama3 8B | ✗ | ✗ | 84.8 | 57.5 | 68.1 | 58.7 |
| | Llama3.1 70B | ✗ | ✗ | 90.3 | 58.2 | 76.8 | 69.3 |

Table 4: Experiment result on certain/uncertain questions with various chunk strategy.

| Chunking | TriviaQA | Natural Question | AmbigNQ | 2WikiQA |
|---|---|---|---|---|
| 512/64 | 90.4/43.0 | 63.7/36.0 | 75.1/33.2 | 66.9/40.6 |
| 256/32 | 88.3/47.7 | 57.6/33.6 | 68.3/33.2 | 61.1/41.7 |
| 128/16 | 87.7/48.9 | 53.6/33.2 | 53.6/31.7 | 58.8/41.4 |
| Full semantic | 87.6/53.0 | 59.3/41.3 | 68.2/39.5 | 65.9/41.5 |
| Adaptive (ours) | 90.3/52.4 | 63.4/41.3 | 75.0/40.5 | 66.9/42.3 |

Table 5: Experiment result of iterative context refinement on uncertain questions.

| Refinement | TriviaQA | Natural Question | AmbigNQ | 2WikiQA |
|---|---|---|---|---|
| w/o refine | 52.4 | 41.3 | 40.5 | 42.3 |
| only removing chunks | 54.5 | 40.6 | 39.5 | 43.3 |
| w/o self-evaluation | 52.7 | 41.5 | 38.2 | 42.1 |
| refill on certain chunks | 55.7 | 41.8 | 43.6 | 44.6 |
| Context-refiner | 56.3 | 42.2 | 43.6 | 46.6 |

**Semantic chunking works on uncertain questions.** Semantic chunking improves uncertain questions but degrades certain ones, consistent with prior findings Qu et al. (2024). For questions already well-answered, semantic chunking restricts information diversity and limits performance. Our adaptive method applies semantic merging only when DENSE indicates high uncertainty, thereby improving uncertain question performance while preserving the advantages of fixed-size chunking on certain ones. Beyond the empirical gains, this observation provides an uncertainty-based explanation for the controversial effectiveness of semantic chunking reported in previous work.

### 6.4.2 ANALYSIS OF ITERATIVE CONTEXT REFINEMENT

We further analyze iterative context refinement through the following settings: (1) no refinement after DENSE, (2) only removing unnecessary chunks, (3) disabling the self-evaluation condition, and (4) refilling based on certain chunks instead of necessary chunks. Table 5 summarizes the results.

Notably, removing unnecessary chunks outperforms the baseline (w/o refine) on TriviaQA and 2WikiQA, with only a minor drop (1%) on Natural Questions and AmbigNQ. This confirms that DENSE effectively identifies and filters irrelevant documents. Disabling self-evaluation leads to consistent drops, showing its usefulness in preventing contexts from generating consistently incorrect responses. Refilling based on certain chunks performs second-best, suggesting that adding information similar to certain chunks can indeed improve QA performance, but the gains are limited compared to refilling guided by necessary chunks.

## 7 CONCLUSIONS

In this work, we explore a fundamental problem in RAG: how to assess whether LLMs understand the retrieved context. We introduced DENSE, a training-free and model-agnostic method that quantifies semantic uncertainty through responses generated under semantically equivalent contexts. Our analysis shows that higher DENSE values consistently correspond to worse performance, validating its effectiveness as an unsupervised measure of contextual understanding. Building on this insight, we designed two modules—Adaptive Semantic Chunking and Iterative Context Refinement—to enhance both intra-chunk semantic coherence and inter-chunk semantic completeness for uncertain questions. Extensive experiments across multiple datasets and backbones demonstrate that DENSE-RAG delivers competitive or superior performance compared to state-of-the-art methods, while requiring no additional training.

Future work could explore adaptive integration of smaller models for simpler tasks to reduce inference costs. Beyond our method on improving context quality, another promising direction for future work is to enhance LLMs' ability to interpret retrieved texts, for example by incorporating uncertainty-aware training objectives during pretraining or finetuning, which may further strengthen QA performance.

**Ethics statement.** Our work focuses on contributions to Retrieval-Augmented Generation (RAG) and does not involve human subjects, private data, or personally identifiable information. All experiments are conducted on publicly available open-book QA datasets, following their respective licenses and intended use.

**Reproducibility statement.** We have made extensive efforts to ensure the reproducibility of our work. The full implementation of our methods, along with detailed instructions for running the experiments, is provided in the anonymous link as well as uploaded Supplementary Materials.

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

APPENDIX

## A THE USE OF LARGE LANGUAGE MODELS (LLMS)

During the preparation of this manuscript, we employed tools such as GPT and Grammarly for language polishing and for assisting in literature search. We emphasize that no part of this work relies on unverified or irresponsible LLM-generated content, and the authors take full responsibility for all contents of the paper.

## B PROMPT FORMATS

### B.1 REPHRASING PROMPT

The format template of LLM inputs in building semantically equivalent contexts as follows:

```
User: Rewrite the following text at the syntactic level without
changing its meaning. Modify the sentence structure, but preserve
the original intent and semantic meaning. ONLY return the
rewritten content without any additional token. Text: {chunk}

LLM: ...
```

### B.2 NLI PROMPT

The format template of LLM inputs in evaluating semantic entailment as follows:

```
User: We are evaluating answers to the question {question} Here
are two possible answers:

Possible Answer 1: {response1}

Possible Answer 2: {response2}

Does Possible Answer 1 semantically entail Possible Answer 2?
Respond with ONLY entailment, contradiction, or neutral.

LLM: ...
```

### B.3 SELF-EVALUATION PROMPT

The format template of LLM inputs in performing self-evaluation on context as follows:

```
User: Context: {context}

Question: {question}

Does the context contain enough information to answer the
question? Only answer yes or no.

LLM: ...
```

### B.4 QA PROMPT

The format template of LLM inputs in asking questions as follows:

```
User: Answer question {query} based on provided context, ONLY
output a short answer with minimum words. Context:{context}

LLM: ...
```

Table 6: Semantic entailment check of rephrased chunks and original ones. 1000 queries are sampled randomly per dataset, and 10 random chunks from its corresponding document are sampled for semantic entailment check.

| Entailment methods | Semantic entailment ratio | | | |
|---|---|---|---|---|
| | TriviaQA | Natural Question | AmbigNQ | 2WikiQA |
| *LLM-as-a-judge* | 98.37% | 95.95% | 97.88% | 93.24% |
| *Deberta-as-a-judge* | 97.65% | 98.28% | 96.35% | 98.40% |

## C    DISENTANGLEMENT OF INHERENT LLM RANDOMNESS

In previous works, uncertainty related metrics are employed under high temperature settings Kuhn et al. (2023); Lin et al. (2024). These works focus on quantifying the intrinsic stochasticity of LLMs as well as their hallucination behavior during question answering. In contrast, we adopt greedy decoding to disentangle the influence of LLM-intrinsic randomness on response variability. To prove the disentanglement works, we run experiment using Llama3-8B on TriviaQA, NQ, AmbigNQ and 2WiKiQA under greedy decoding and fixed context. We then compute discrete semantic entropy using the official implementation provided by Kuhn et al. (2023). In all datasets, the measured uncertainty is exactly zero, confirming that our greedy decoding setup successfully eliminates randomness-induced variability from the LLM itself. And the observed uncertainty in DENSE is originated from rephrased contexts. Lin et al. (2024) explored the relationship between decoding temperature and uncertainty estimation. For more details on this topic, we refer readers to their work.

## D    REPHRASED CONTEXTS SEMANTIC CHECKING

To validate that the LLM rephrasing in DENSE could preserve the semantic of the original chunks, we perform semantic checking of rephrased chunks on four QA dataset that we used in main experiments. We randomly sample 1000 questions in each QA dataset, and for each question we randomly sample 10 chunks in its corresponding retrieved documents. We employ both LLM and Deberta-large-mnli model to decide whether rephrased chunks are semantically entailed with original chunks. The result is shown in Table 6, which indicating that the rephrased chunks shows negligible semantic shift compare to original chunks. This proves that the rephrasing operation have negligible influence on QA performance.

## E    DISCUSSION OF SEMANTIC ENTROPY AND DENSE

Kuhn et al. (2023) define the semantic entropy to measure the uncertainty of LLM's responses:

$$SE(q, C) \approx -|H|^{-1} \sum_{i=1}^{|H|} \log p(h|C), \tag{11}$$

where $h$ is a semantic cluster belongs to $H = \{h_1, h_2...\}$, $p(h|x)$ estimates a categorical distribution over the cluster meanings. The cluster is computed by bi-directional entailment $E_{r_i, r_j}$. A natural language inference model (NLI) is used to classify whether $r_i$ and $r_j$ are *entailment(1)* or *neutral(0)*:

$$E_{r_i, r_j} = (NLI(r_i, r_j) + NLI(r_j, r_i))/2, \tag{12}$$

a Deberta-large model is employed to make the classification. $r_i$ and $r_j$ are clustered together into one semantic cluster $h$ when $E_{r_i, r_j} = 1$.

We discuss the relationship of semantic entropy and the degree-based semantic entropy in DENSE from three perspectives:

*Monotonicity*: Both formulations decrease monotonically as semantic consistency among responses increases. In semantic entropy, diverse semantic lead to more clusters, and $p(H|x)$ decrease, the entropy increase. In our method, diverse semantic lead to smaller $D$, the entropy also increase.

*Range Analysis*: We now show that the two formulations share the same value range by analyzing two extreme cases. In the ideal case, where all responses are semantically equivalent, and there will

be only one cluster in equation 11:

$$SE(x) \approx -1^{-1} \log 1 = 0 \tag{13}$$

And in our method, the $D$ of all response will be $k + 1$:

$$DSE(x) \approx -(k+1)^{-1} \sum_{i=0}^{k+1} \log 1 = 0 \tag{14}$$

In the ideal case, two equations are equivalent. In the worst case, where all response shows distinct semantic, there are $k + 1$ clusters in equation 11 and each $p(C_i|x) = \frac{1}{k+1}$:

$$SE(x) \approx -(k+1)^{-1} \sum_{i=1}^{k+1} \log \frac{1}{k+1} = -(k+1)^{-1}(k+1)log(k+1)^{-1} = \log(k+1) \tag{15}$$

In our method, the $D = 1$ for all response, since every response only semantically equal to itself:

$$DSE(x) \approx -(k+1)^{-1} \sum_{i=1}^{k+1} \log \frac{1}{k+1} = \log(k+1) \tag{16}$$

Hence, two methods have a value range of $[0, log(k+1)]$, and have the same value of both ideal case and the worst case.

*Information-Theoretic Interpretation.* From an information-theoretic perspective, semantic uncertainty is derived from Shannon entropy over the distribution $\{p(h_i|x)\}$, where $h_i$ denotes a semantic cluster of responses. Intuitively, $p(h_i|x)$ represents the probability that a randomly sampled response falls into cluster $h_i$.

Our method avoids the explicit clustering step by directly considering the semantic graph, where each node is a response and edge weights represent pairwise semantic entailment. The degree $D_i$ measures how semantically connected a response is to all others — in other words, it approximates how many responses are semantically similar to $r_i$.

## F    Algorithms

### F.1    DENSE

We present the DENSE algorithm in Algorithm 1.

---

**Algorithm 1:** DENSE

---

**Input:** Chunk set $\{c_1, c_2, c_3, \ldots, c_k\}$, question $q$, LLM $f_\theta$, rephrase prompt $p_r$, QA prompt $p_q$, NLI agent

**Output:** degree-based semantic entropy $DSE$, reply set $R$, semantic matrix $W$

1  $R \leftarrow \emptyset$
2  $W \leftarrow [0]_{k+1 \times k+1}$
3  **foreach** $c_i \in \{c_1, c_2, \ldots, c_k\}$ **do**
4  $\quad c_i' \leftarrow f_\theta(p_r(c_i))$
5  **for** $i \leftarrow 1$ **to** $k$ **do**
6  $\quad C_i \leftarrow [c_1, \ldots, c_i', \ldots, c_k]$
7  $\quad r_i \leftarrow f_\theta(p_q(q, C_i))$
8  $\quad R \leftarrow R \cup \{r_i\}$
9  **foreach** $r_i \in R$ **do**
10 $\quad$ **foreach** $r_j \in R$ **do**
11 $\quad\quad w_{ij} \leftarrow (NLI(r_i, r_j) + NLI(r_j, r_i))/2$
12 $\quad\quad W[i, j] \leftarrow w_{ij}$
13 $DSE \leftarrow -(k+1)^{-1} \sum_{i=1}^{k+1} \log \frac{D_i}{k+1}, \; where \; D_i = \sum_{j=0}^{k} w_{ij}$
14 **return** $DSE, R, W$

---

**Algorithm 2:** Adaptive Semantic Chunking
___
**Input:** Document set $\mathcal{D} = \{d_1, d_2, \ldots, d_n\}$, Similarity threshold $\tau$, Maximum chunk length $T_{max}$
**Output:** Semantic chunk set $\mathcal{C}$
1 Initialize an empty list: $\mathcal{C} \leftarrow \emptyset$
2 **foreach** $d \in \mathcal{D}$ **do**
3    $S \leftarrow \texttt{tokenize}(d)$ // split documents into sentences
4    $\{e_1, e_2, e_3, \ldots\} \leftarrow [\texttt{Encode}(s) \mid s \in S]$ // encode sentences into embeddings
5    $c \leftarrow [s_1]$ // initialize current chunk
6    $T_{current} \leftarrow |s_1|$
7    **for** $i \leftarrow 1$ **to** $|S| - 1$ **do**
8      $sim \leftarrow \texttt{cosine\_similarity}(e_i, e_{i+1})$
9      $T_{next} \leftarrow |s_{i+1}|$
10      **if** $sim > \tau$ **and** $T_{current} + T_{next} \leq T_{max}$ **then**
11        $c \leftarrow c \cup \{s_{i+1}\}$
12        $T_{current} \leftarrow T_{current} + T_{next}$
13      **else**
14        $\mathcal{C} \leftarrow \mathcal{C} \cup \{c\}$
15        $c \leftarrow [s_{i+1}]$ // initialize a new chunk
16        $T_{current} \leftarrow T_{next}$
17    **if** $c$ *is not empty* **then**
18      $\mathcal{C} \leftarrow \mathcal{C} \cup \{c\}$
19 **return** $\mathcal{C}$

### F.2 ADAPTIVE SEMANTIC CHUNKING

We present the adaptive semantic chunking Algorithm 2.

### F.3 ITERATIVE CONTEXT REFINEMENT

We present the Iterative Context Refinement algorithm in Algorithm 3.

## G DATASETS

We describe the open-book QA dataset here. Since all proposed methods requires no training, we only use the *dev* sets for evaluation. The statistics of each dataset is shown in Table 7:

- **TriviaQA** Joshi et al. (2017) is a challenging QA dataset that provinding evidence documents. There are two types of questions: Wikipedia and Web. We follow KILT benchmark, only consider Wikipedia cases Petroni et al. (2020) with evidence documents. We use the *wikipedia-dev* set in experiments.

- **Natural Question** Kwiatkowski et al. (2019) is a common-used QA dataset, which is extracted from Wikipedia. The questions are constructed from Google search engine and the provided documents are corresponding Wikipedia pages. We follow KILT benchmark Petroni et al. (2020) and only consider questions for which at least one human annotator has marked a short answer in the documents. We only use the *dev* set in experiments.

- **AmbigNQ** Min et al. (2020) is a QA dataset proposed in AmbigQA, which is constructed using prompt questions from NQ-OPEN and English Wikipedia as the evidence corpus. In our task, we consider the *singleAnswer* questions in *dev* subset in AmbigNQ.

- **2WikiQA** Ho et al. (2020) is a multi-hop QA dataset, which is designed to test the relationship between two entities. In 2WikiQA, multiple evidence articles are provided for one question. We use the *dev* set in our experiments.

---

**Algorithm 3:** Iterative Context Refinement

---

**Input:** Question $q$, $Chunk\ Set$, $k$, LLM $f_\theta$, self evaluation prompt $p_e$,
**Output:** $Answer_{best}$

**1 Initialization:**
**2** $C \leftarrow$ initial top-$k$ chunks for $q$
**3** $DSE_{min} \leftarrow \infty, Flag_{current} \leftarrow 0$
**4 while** $Visited\ Set \neq Chunk\ Set$ **do**
**5**     $Visited\ Set \leftarrow Visisted\ Set \cup C$
**6**     $DSE, R, W \leftarrow DENSE(q, C)$
**7**     $Flag_{eval} \leftarrow f_\theta(p_e(q, C))$
**8**     **if** $DSE < DSE_{min}$ **then**
**9**        **if** $Flag_{current} = 0 \lor Flag_{eval} = 1$ **then**
**10**           $Answer_{best} \leftarrow R$
**11**           $DSE_{min} \leftarrow DSE$
**12**           $Flag_{current} = Flag_{eval}$
**13**     **if** $DSE_{min} < 0.2$ & $Flag_{current} = 1$ **then**
**14**        **Return:** $Answer_{best}$;
**15**     $Visited\ Set \leftarrow \emptyset, Certain\ Set \leftarrow \emptyset, Uncertain\ Set \leftarrow \emptyset, Necessary\ Set \leftarrow \emptyset,$
      $Unnecessary\ Set \leftarrow \emptyset$
**16**     **for** $i \leftarrow 1$ **to** $k$ **do**
**17**        **if** $W_{i0} = 1$ & $W_{0i} = 1$ **then**
**18**           $Certain\ Set \leftarrow Certain\ Set \cup \{c_i\}$;
**19**        **else**
**20**           $Uncertain\ Set \leftarrow Uncertain\ Set \cup \{c_i\}$;
**21**     **for** $c_i \in Uncertain\ Set$ **do**
**22**        $AC_i \leftarrow [c_0, ..., c_{i-1}, c_{i+1}, ...]$
**23**        $ar_i \leftarrow f_\theta(q, AC_i)$
**24**        **if** $E(r_0, ar_i) = 1$ **then**
**25**           $Unnecessary\ Set \leftarrow Unnecessary\ Set \cup \{c_i\}$
**26**        **else**
**27**           $Necessary\ Set \leftarrow Necessary\ Set \cup \{c_i\}$
**28**     **for** $c_i$ **in** $Unnecessary\ Set$ **do**
**29**        $C \leftarrow C \setminus \{c_i\}$;
**30**     **for** $c_i$ **in** $Necessary\ Set$ **do**
**31**        $max\_sim \leftarrow -\infty$
**32**        $best\_chunk \leftarrow \emptyset$
**33**        **for** $c_j$ **in** $Chunk\ Set \setminus Visited\ Set$ **do**
**34**           $sim \leftarrow cosine\_similarity(c_i, c_j)$;
**35**           **if** $sim > max\_sim$ **then**
**36**              $max\_sim \leftarrow sim\_score$;
**37**              $best\_chunk \leftarrow c_j$;
**38**        $C \leftarrow C \cup \{best\_chunk\}, Visited\ Set \leftarrow Visited\ Set \cup best\_chunk$
**39 Return:** $Answer_{best}$

---

Table 7: Dataset Statistics. The certain and uncertain questions are devided by DENSE in Llama-3.1 8B.

| Datasets | No. all valid questions | No. certain questions | No. uncertain questions |
|---|---|---|---|
| TriviaQA | 7928 | 6726 | 1202 |
| NQ | 4289 | 3115 | 1174 |
| AmbigNQ | 1000 | 780 | 220 |
| 2WikiQA | 12576 | 7663 | 4913 |

## H  IMPLEMENTATION DETAILS

We implement a naive RAG framework Gao et al. (2023) on Qwen-2.5 1.5B, Qwen-3 8B, Llama-3 8B, Llama-3.1 8B and Llama-3.1 70B as backbones. We use the documents provided within dataset

Table 8: AUROC and AURAC under high-temperature sampling. LLM is no longer faithfully grounded in the retrieved evidence in high-temperature sampling, uncertainty scores in this regime lose their interpretability for retrieval-based QA.

| Uncertainty Measurements | TriviaQA | | Natural Question | | AmbigNQ | | 2WikiQA | |
|---|---|---|---|---|---|---|---|---|
| | AUROC | AUARC | AUROC | AUARC | AUROC | AUARC | AUROC | AUARC |
| *temperature=1.0* | | | | | | | | |
| Discrete Semantic Entropy Farquhar et al. (2024) | 78.82 | 93.23 | 72.29 | 79.15 | 72.19 | 82.67 | 67.18 | 67.61 |
| $U_{eigv}$ Lin et al. (2024) | **81.75** | 93.87 | **75.12** | 81.49 | **75.50** | 82.90 | 68.01 | 68.12 |
| $U_{deg}$ Lin et al. (2024) | 81.69 | 93.78 | 75.00 | 81.35 | 74.98 | **83.25** | 67.78 | 67.86 |
| KLE heat$_{t=0.1}$ Nikitin et al. (2024) | 81.30 | 93.72 | 74.84 | **81.65** | 74.31 | 82.50 | 67.29 | 67.67 |
| KLE deberta matern$_{\kappa=3.0,\nu=3.0}$ Nikitin et al. (2024) | 71.62 | 91.78 | 67.73 | 77.86 | 66.18 | 79.47 | 62.86 | 66.28 |
| DENSE$_{llm}$(ours) | 80.20 | **93.92** | 70.92 | 77.00 | 74.73 | 81.68 | **70.05** | **70.25** |
| *temperature=3.0* | | | | | | | | |
| Discrete Semantic Entropy Farquhar et al. (2024) | 84.11 | 90.60 | **77.36** | **69.78** | 80.23 | 79.15 | 73.15 | 56.86 |
| $U_{eigv}$ Lin et al. (2024) | 80.42 | 89.55 | 67.35 | 64.45 | 75.73 | 76.91 | 70.06 | 55.50 |
| $U_{deg}$ Lin et al. (2024) | 84.00 | 90.87 | 72.45 | 67.63 | 79.81 | 79.12 | 72.51 | 57.21 |
| KLE heat$_{t=0.1}$ Nikitin et al. (2024) | **85.74** | **91.44** | 75.31 | 69.40 | **81.72** | **80.08** | **73.89** | **58.03** |
| KLE deberta matern$_{\kappa=3.0,\nu=3.0}$ Nikitin et al. (2024) | 59.95 | 76.50 | 58.22 | 58.16 | 58.34 | 66.54 | 56.16 | 45.01 |
| DENSE$_{llm}$(ours) | 78.58 | 87.33 | 72.33 | 65.14 | 75.39 | 71.07 | 70.33 | 53.93 |
| *temperature=5.0* | | | | | | | | |
| Discrete Semantic Entropy Farquhar et al. (2024) | **68.08** | **49.21** | 61.22 | 20.51 | 62.74 | 24.73 | 71.31 | 21.45 |
| $U_{eigv}$ Lin et al. (2024) | 65.06 | 47.14 | 56.51 | 18.19 | 60.62 | 23.32 | 71.93 | 22.05 |
| $U_{deg}$ Lin et al. (2024) | 66.53 | 48.20 | 58.07 | 18.89 | 62.03 | 24.20 | 72.30 | 22.30 |
| KLE heat$_{t=0.1}$ Nikitin et al. (2024) | 67.88 | 48.96 | 59.31 | 19.48 | 63.28 | 24.84 | **72.33** | 22.24 |
| KLE deberta matern$_{\kappa=3.0,\nu=3.0}$ Nikitin et al. (2024) | 51.15 | 34.49 | 48.02 | 14.74 | 49.13 | 18.44 | 51.92 | 12.33 |
| DENSE$_{llm}$(ours) | 66.94 | 46.45 | **62.25** | **20.99** | **68.96** | **27.69** | 71.10 | **22.53** |
| *temperature=7.0* | | | | | | | | |
| Discrete Semantic Entropy Farquhar et al. (2024) | 64.67 | 36.77 | 59.68 | 13.86 | 63.77 | 18.77 | 68.32 | 16.89 |
| $U_{eigv}$ Lin et al. (2024) | 64.31 | 36.27 | 57.58 | 12.91 | 62.35 | 18.29 | 71.12 | 18.38 |
| $U_{deg}$ Lin et al. (2024) | 64.90 | 36.66 | 58.30 | 13.30 | 63.92 | 18.66 | 70.94 | 18.31 |
| KLE heat$_{t=0.1}$ Nikitin et al. (2024) | 65.51 | 36.81 | 58.73 | 13.51 | 64.16 | **18.78** | 70.23 | 17.97 |
| KLE deberta matern$_{\kappa=3.0,\nu=3.0}$ Nikitin et al. (2024) | 52.33 | 27.63 | 49.98 | 10.88 | 49.47 | 12.23 | 52.94 | 10.83 |
| DENSE$_{llm}$(ours) | **66.43** | **38.83** | **61.91** | **15.22** | **66.13** | 18.04 | **74.03** | **20.76** |

as the retrieval corpus and employ recursive chunking in Langchain[1]. The chunk size is set to 512 and chunk overlap is set to 64. For document retrieval, we use UAE-Large-V1 as the encoder for both questions and document chunks, which is one of the best zero-shot embedding models in MTEB (eng, v2) leaderboard[2]. Then we employ FAISS[3] to build dense index. To ensure a fair comparison with other baselines and demonstrate that the improvements of our method stem from better document understanding rather than an increased number of documents, we limit the retrieval to the top 5 documents—consistent with the minimum retrieval setting used in most RAG studies. We discuss the impact of different values of top-$k$ on the model's performance in Appendix K. In generation stage, we use a simple prompt which is described in Appendix B.4.

In adaptive semantic chunking, we set the merge threshold $\tau = 0.6$ and use the same encoder in embeddin chunks as the sentence encoder. The maximum chunk length $T_{max} = 512$, consistent with recursive chunking. The NLI agent and the LLM for semantically equivalent context construction are the LLM used in generation stage. For the 1.5B and 8B DENSE-RAG, a single NVIDIA 3090 GPU is enough for embedding and inference. For the 70B DENSE-RAG, we use 2 NVIDIA A100 80GB GPUs for embedding and inference.

# I    MORE DENSE EVALUATION EXPERIMENTAL RESULTS

Here we show the performance of RAG on certain and uncertain questions when using different LLM in open-book QA in Table 9. Regardless of model size, all LLMs exhibit a significant performance drop on uncertain questions. While the drop is mitigated for larger models such as the llama3.1-70B, it still remains around 20%. This demonstrates that DENSE generalizes well across language models of different scales.

## I.1    EXPERIMENTS IN HIGH-TEMPERATURE SAMPLING

In addition to the low-temperature and greedy decoding evaluations reported in the main paper, we further examine the behavior of DENSE and other uncertainty estimating baselines under higher sampling temperatures. The result are shown in Table 8. Also we plot the QA performance changes as temperature increasing in Figure 6. As temperature increases, the generation process becomes increasingly dominated by stochastic variations within the LLM rather than by differences in how

---

[1]https://python.langchain.com/docs/concepts/text_splitters/

[2]https://huggingface.co/spaces/mteb/leaderboard

[3]https://ai.meta.com/tools/faiss/

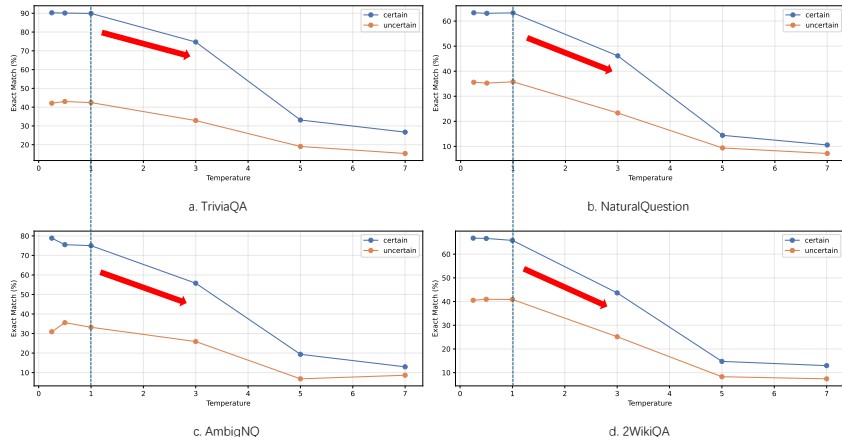

Figure 6: RAG QA performance in different sampling temperature. Once the temperature exceeds 1, the exact match accuracy drops sharply, indicating that the model becomes increasingly unfaithful to the retrieved context.

Table 9: RAG performance on **certain** and **uncertain** questions. The EM% drop on uncertain questions to certain one is reported after the EM on ucnertain question. The split of certain questions and uncertain questions is according to DENSE$_{eval}$.

| Task | TriviaQA | | Natural Question | | AmbigNQ | | 2WikiQA | |
|---|---|---|---|---|---|---|---|---|
| | certain | uncertain | certain | uncertain | certain | uncertain | certain | uncertain |
| Llama3.1 8B DENSE$_{eval}$ | 90.4 | 43.0(47.4↓) | 63.7 | 36.0(26.7↓) | 75.1 | 33.2(41.9↓) | 66.9 | 40.6(26.3↓) |
| Llama3.1 8B DENSE$_{deberta}$ | 90.3 | 43.7 (46.6↓) | 63.8 | 36.0(27.8↓) | 75.1 | 34.1(41.0↓) | 67.0 | 41.2(25.8↓) |
| Qwen2.5 1.5B | 75.0 | 33.3(41.7↓) | 47.7 | 24.8(22.9↓) | 55.5 | 21.8(33.7↓) | 38.5 | 21.4(17.1↓) |
| Llama3.1 8B | 90.3 | 42.3(48.0↓) | 63.3 | 35.8(27.5↓) | 75.6 | 32.8(42.8↓) | 66.9 | 40.3(26.6↓) |
| Llama3.1 70B | 93.2 | 73.1(20.1↓) | 62.5 | 44.8(17.7↓) | 81.8 | 59.6(22.2↓) | 72.3 | 57.4(14.9↓) |

the retrieved context is interpreted. As a result, the semantic deviations in LLM outputs no longer reliably reflect context-related uncertainty but instead arise primarily from temperature-induced randomness. Consistent with this shift, all methods—DENSE included—degrades substantially at high temperatures. Because LLM is no longer faithfully grounded in the retrieved evidence, uncertainty scores in this regime lose their interpretability for retrieval-based QA. We therefore include these results only for completeness; they should not be taken as indicators of method quality in retrieval-grounded settings.

## I.2 EXPERIMENTS ON SUMMARIZATION TASKS

We focuses on the open-book QA task, but as a typical free-form language generation task, we also explore its effectiveness on summarization tasks. We add a simple verification experiment on CNN/DailyMail 3.0.0 test set. We build a RAG pipeline for summarization and compute the DENSE score, the RougeL and DENSE score relationship is shown in Table 10. As shown in the figure, higher uncertainty is correlated with lower RougeL score, showing DENSE's potential in measuring LLM's understanding in summarization tasks. This verification experiment on CNN/DailyMail is preliminary, as we directly applied the method originally designed for open-book QA. Summarization presents different challenges compared to open-book QA, such as how to formulate effective queries. One important direction is to improve summarization performance according to the proposed DENSE method.

Table 10: Experiment result on CNN/DailyMail summarization.

| DENSE range | ( ,0.1) | [0.1, 0.2) | [0.2, 0.3) | [0.3, 0.4) | [0.4, 0.5) | [0.5, 0.6) | [0.6, 0.7) | [0.7, 0.8) | [0.8, 0.9) | [0.9, 1.0) | [1.0, ) |
|---|---|---|---|---|---|---|---|---|---|---|---|
| Ave RougeL/$10^{-3}$ | 90.47 | 86.97 | 86.84 | 82.40 | 86.29 | 81.48 | 78.56 | 76.54 | 80.36 | 81.48 | 76.43 |

Table 11: Results of our methods and baselines on 4 datasets. The best results are in **bold**, and the second best are underlined. Results unavailable in public reports are marked as "-".

| Method | Generator Model | Ret. FT | Gen. FT | TriviaQA | Natural Question | AmbigNQ | 2WikiQA |
|---|---|---|---|---|---|---|---|
| Adaptive-RAGJeong et al. (2024) | FLAN-T5-XL 3B | ✓ | ✗ | 52.2 | 37.8 | - | 40.6 |
| Astute-RAG Wang et al. (2024b) | Claude 3.5 Sonnet | ✗ | ✗ | 84.5 | 53.6 | - | - |
| RePlug-LSR (few-shot)Shi et al. (2023) | Codex 175B | ✓ | ✗ | 77.3 | 45.5 | - | - |
| LongRAGZhao et al. (2024) | GLM4 32B | ✗ | ✓ | - | - | - | 57.2 |
| ERM4Shi et al. (2024) | GPT-3.5-turbo | ✓ | ✗ | - | 52.7 | 53.5 | 46.8 |
| UncertaintyRAGLi et al. (2024b) | Vicuna 7B | ✓ | ✗ | 85.0 | - | - | 29.9 |
| UncertaintyRAGLi et al. (2024b) | Llama2 13B | ✓ | ✗ | 82.5 | - | - | 38.3 |
| RAG-DDRLi et al. (2024a) | Llama3 8B | ✓ | ✓ | 89.6 | 52.1 | - | - |
| ChatQA-1.5Liu et al. (2024b) | Llama3 8B | ✓ | ✓ | 81.0 | 42.4 | - | 26.8 |
| ChatQA-1.5Liu et al. (2024b) | Llama3 70B | ✓ | ✓ | 85.6 | 47.0 | - | 34.9 |
| RankRAGYu et al. (2024) | Llama3 8B | ✗ | ✓ | 82.9 | 50.6 | - | 31.4 |
| RankRAGYu et al. (2024) | Llama3 70B | ✗ | ✓ | 86.5 | 54.2 | - | 38.2 |
| Collab-RAGXu et al. (2025) | Qwen2.5 3B | ✗ | ✓ | - | - | - | 67.0 |
| Collab-RAGXu et al. (2025) | Llama3.1 8B | ✗ | ✓ | - | - | - | 67.2 |
| | Qwen2.5 1.5B | ✗ | ✗ | 69.8 | 42.4 | 50.3 | 31.5 |
| | Qwen3 8B | ✗ | ✗ | 82.4 | 53.5 | 65.2 | 54.2 |
| DENSE-RAG | Llama3 8B | ✗ | ✗ | 84.8 | 57.5 | 68.1 | 58.7 |
| | Llama3.1 8B | ✗ | ✗ | 85.1 | 57.8 | 67.9 | 59.1 |
| | Llama3.1 70B | ✗ | ✗ | **90.3** | **58.2** | **76.8** | **69.3** |

## J  COMPARE WITH MORE BASELINES

We show extended comparison in Table 11. We consider following sota RAG methods: Astute RAG Wang et al. (2024b), RePlug Shi et al. (2023), Adaptive-RAG Jeong et al. (2024), shi et al. Shi et al. (2024), LongRAG Zhao et al. (2024), RAG-DDR Li et al. (2024a), ChatQA-1.5 Liu et al. (2024b), RankRAG Yu et al. (2024) and UncertaintyRAG Li et al. (2024b). Among these baselines, some baselines like Yu et al. (2024) employ finetuned LLMs to further optimize the retrieved context; in the table, we mark them simply as Gen. FT. Only approaches that introduce additional trained components such as retrievers, encoders, or policy models, are regarded as using trained retrievers. For Collab-RAG Xu et al. (2025), it utilize GPT4o as an LLM reader during the retrieval and generation. It is worth noting that some baselines employ different retrieval settings, such as retrieving a larger number of documents or searching over Wikipedia rather than the dataset-provided corpus; their results are thus reported for reference only. For fair comparison, our method uniformly uses the top-5 retrieved chunks (the minimum number adopted in most prior work) as context and performs retrieval strictly over the dataset-provided knowledge base.

## K  ADDITIONAL SENSITIVITY ANALYSIS

**Number of retrieved documents.** To discuss the performance of DENSE with different numbers of retrieved chunks, we conduct experiments with different chunk numbers with and without DENSE. As shown in Figure 8, when more chunks are utilized, two methods have better performance on uncertain questions. And DENSE-RAG consistently outperforms RAG w/o DENSE across all datasets and various chunk quantity settings, highlighting the robustness of our approach.

**Adaptive chunking threshold** $\tau$ We run experiments with different adaptive chunking threshold and show the result in Figure 7. As the sentence merge threshold of adaptive chunking increases, performance on uncertain questions significantly declines. This indicates that while leveraging semantic similarity for chunking can enhance performance, overly strict merge conditions may instead lead to a drop in overall effectiveness.

## L  DISCUSSION OF COMPLEXITY

As an unsupervised method applicable to any black-box LLM without introducing additional models, DENSE requires multiple model calls during measurement. The computational complexity remains bounded by $O(k^2)$, where each LLM inference step (e.g., response generation or entailment check) is treated as $O(1)$. The main steps include: (1) rephrasing $k$ retrieved chunks ($O(k)$); (2) generating responses under rephrased contexts ($O(k)$); and (3) pairwise entailment comparisons ($O(k^2)$). Notably, the entailment component can be efficiently accelerated using lightweight NLI models such as

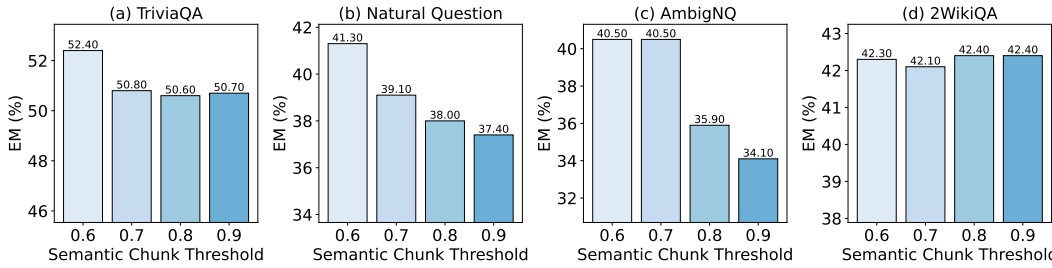

Figure 7: Performance comparison on uncertain questions when using different Adaptive chunking merging thresholds.

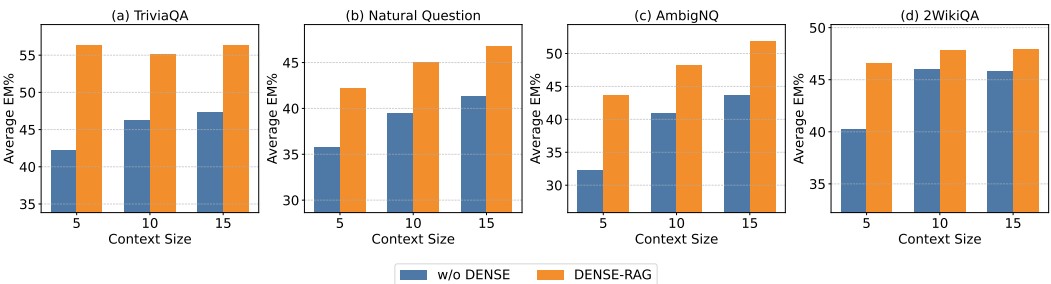

Figure 8: Performance comparison on uncertain questions when using different context size. Under different context sizes, DENSE-RAG demonstrates significant improvements.

DeBERTa Kuhn et al. (2023), instead of relying on repeated LLM calls. Compared with DSE Kuhn et al. (2023), the only additional cost introduced by DENSE is the rephrasing step.

For semantic chunking, we embed sentences using a compact encoder rather than an LLM, so the overhead is negligible relative to the $O(k^2)$ entailment computations. In the refinement stage, the worst-case scenario occurs when all candidate chunks must be examined. This introduces up to $O(k)$ additional generations and $O(k^2)$ additional DENSE computations, resulting in an overall upper bound of $O(k^2)$ complexity. However, in our experiments, only about 20% of the questions exhibit sufficiently high DENSE scores to trigger refinement. Therefore, the practical computational cost remains close to $O(k^2)$ while being incurred only for a small subset of difficult queries.

In practice, on Llama-3 with A100 GPUs, computing DENSE adds about one second per query, while the full DENSE-RAG pipeline averages six seconds. This additional cost remains modest compared to standard RAG inference and represents a highly favorable trade-off, as it enables reliable, training-free uncertainty measurement and effective context enhancement in a fully model-agnostic and widely generalizable manner.

## M CASE STUDIES

We display the case studies of DENSE-RAG on uncertain questions in TriviaQA, Natural Question, AmbigNQ and 2WikiQA. We display the retrieved chunks and LLM responses with/without DENSE in Table 12, Table 13, Table 14 and Table 15. The red text is distractors, and the green text contains evidence for the correct answer. From all the cases, we can observe that these uncertain questions are often accompanied by distracting texts, which can easily mislead the LLM. Take 2WikiQA as an example: LLM needs to first identify the author of Sunday Papers and then locate the awards won by that author. By using DENSE, we effectively identify the chunks that are relevant but do not contain the answer, allowing the chunk refiner to pinpoint the chunk that holds the correct information.

Table 12: Case study on uncertain questions in TriviaQA.

| Question Id: tc_1693 |
| --- |
| **Q: Who beat Tim Henman in his first Wimbledon singles semifinal?** A: Pete Sampras |

| | |
| --- | --- |
| w/o DENSE | **Chunk 1**: Timothy Henry "Tim" Henman (born 6 September 1974) is a retired English professional tennis player. Henman played a serve-and-volley style of tennis... 
 **Chunk 2**: In the second round he succumbed to the eventual champion American Todd Martin, 6–4, 6–4. Henman received a wildcard for the Manchester Open, where he lost in the first round to American Alex O'Brien... 
 **Chunk 3**: At the time of his retirement, Henman had already committed to playing a Charity Exhibition at London's Royal Albert Hall during the Seniors Tennis Event The Blackrock Masters in December 2007... 
 **Chunk 4**: Then breaking his opponent's serve twice in a row to win the final set 7–5 and beat reigning French Open champion Yevgeny Kafelnikov in the first round at Wimbledon, going on to reach the quarter finals before losing to Todd Martin... 
 **Chunk 5**: He reached the second round after defeating German Martin Sinner, and in Nottingham he reached the quarter-finals, his first quarter-final in the ATP tour. His success in these tournaments increased his ranking from 272nd to 219th. 
 LLM response: Todd Martin |
| DENSE-RAG | **Chunk 1**: Timothy Henry "Tim" Henman (born 6 September 1974) is a retired English professional tennis player. Henman played a serve-and-volley style of tennis... 
 **Chunk 2**: In 2000 he reached the fourth round and in 1996, 1997, 2003 and 2004 he lost in the quarter-finals. The first two of those semi-final losses were to Pete Sampras, who went on to win the title on both occasions... 
 **Chunk 3**: Then breaking his opponent's serve twice in a row to win the final set 7–5 and beat reigning French Open champion Yevgeny Kafelnikov in the first round at Wimbledon, going on to reach the quarter finals before losing to Todd Martin... 
 **Chunk 4**: On the grass at Queen's Club Championship Henman reached the final, where he was defeated in straight sets by Australian Lleyton Hewitt... 
 **Chunk 5**: However, Henman's winning streak did not last long, and in the second round he met Sampras, and was defeated 6–2, 6–3, 7–6... 
 LLM response: Pete Sampras |

Table 13: Case study on uncertain questions in Natural Question.

| Question Id: -1580637763990313083 |
| --- |
| **Q: When did muhammad ali win an olympic gold medal?** A: 1960 |

| | |
| --- | --- |
| w/o DENSE | **Chunk 1**: Ali said in his 1975 autobiography that shortly after his return from the Rome Olympics, he threw his gold medal into the Ohio River after he and a friend were refused service at a "whites - only" restaurant and fought with a white gang... 
 **Chunk 2**: Ali received a replacement medal at a basketball intermission during the 1996 Olympics in Atlanta, where he lit the torch to start the gam professional boxing early career... 
 **Chunk 3**: On September 13, 1999, Ali was named "Kentucky Athlete of the Century" by the Kentucky Athletic Hall of Fame in ceremonies at the Galt House East. On January 8, 2001, Muhammad Ali was presented with the Presidential Citizens Medal by President Bill Clinton. 
 **Chunk 4**: In 1996, he had the honor of lighting the flame at the 1996 Summer Olympics in Atlanta, Georgia. Ali 's bout with Parkinson 's led to a gradual decline in his health... 
 **Chunk 5**: W. Bush embraces Muhammad Ali after presenting him with the Presidential Medal of Freedom on November 9, 2005, during ceremonies at the White House. Muhammad Ali defeated every top heavyweight in his era, which has been called the golden age of heavyweight boxing... 
 LLM response: Muhammad Ali did not win an Olympic gold medal. |
| DENSE-RAG | **Chunk 1**: On September 13, 1999, Ali was named "Kentucky Athlete of the Century" by the Kentucky Athletic Hall of Fame in ceremonies at the Galt House East... 
 **Chunk 2**: Legacy As Mrs. Lonnie Ali looks on, President George W. Bush embraces Muhammad Ali after presenting him with the Presidential Medal of Freedom on November 9, 2005, during ceremonies at the White House... 
 **Chunk 3**: Muhammad Ali defeated every top heavyweight in his era, which has been called the golden age of heavyweight boxing. Ali was named "Fighter of the Year" by The Ring magazine more times than any other fighter, and was involved in more Ring "Fight of the Year" bouts than any other fighter. 
 **Chunk 4**: At age 18, he won a gold medal in the light heavyweight division at the 1960 Summer Olympics in Rome and turned professional later that year. At age 22 in 1964, he won the WBA, WBC, and lineal heavyweight titles from Sonny Liston in a major upset. 
 **Chunk 5**: In time, Muhammad Ali Boulevard – and Ali himself – came to be well accepted in his hometown. In 1993, the Associated Press reported that Ali was tied with Babe Ruth as the most recognized athlete, out of over 800 dead or living athletes, in America. 
 LLM response: 1960 |

Table 14: Case study on uncertain questions in AmbigNQ.

| Question Id: 3008542401703695479 |
| --- |
| **Q: Who played robin on the original batman series?** A: Burt Ward |

| | |
| --- | --- |
| w/o DENSE | **Chunk 1**: Robin (Dick Grayson) was portrayed by Douglas Croft and Johnny Duncan, respectively, in the 1943 and 1949 fifteen chapter Batman serials. Burt Ward played him in the 1966–1968 Batman television series and the related 1966 film... 
 **Chunk 2**: Batman is a 1960s American live action television series, based on the DC comic book character of the same name. It stars Adam West as Bruce Wayne / Batman and Burt Ward as Dick Grayson / Robin – two crime-fighting heroes who defend Gotham City from a variety of arch villains. 
 **Chunk 3**: Lewis Wilson as Bruce Wayne / Batman, Douglas Croft as Richard "Dick" Grayson / Robin J. Carrol Naish as Dr. Tito Daka/Prince Daka * Shirley Patterson as Linda Page * William Austin as Alfred Pennyworth... 
 **Chunk 4**: Burt Ward as Dick Grayson / Robin: Batman's sidekick, dubbed the "Boy Wonder": a high school student noted for his recurring interjections in the form of "Holy _, Batman!" 
 **Chunk 5**: Lewis Wilson as Batman and Douglas Croft as Robin: Both actors and their stunt doubles lacked the "style and grace" of either the comic characters they were portraying or their equivalents at Republic Pictures. 
 LLM response: Douglas Croft |
| DENSE-RAG | **Chunk 1**: His portrayal is based mainly on Burt Ward's Dick Grayson. Collected editions See also Homosexuality in the Batman franchise * List of exclamations by Robin... 
 **Chunk 2**: Portrayals Robin (Dick Grayson) was portrayed by Douglas Croft and Johnny Duncan, respectively, in the 1943 and 1949 fifteen chapter Batman serials... 
 **Chunk 3**: Batman is a 1960s American live action television series, based on the DC comic book character of the same name. It stars Adam West as Bruce Wayne / Batman and Burt Ward as Dick Grayson / Robin – two crime-fighting heroes who defend Gotham City from a variety of arch villains... 
 **Chunk 4**: The entire first Batman serial was re-released theatrically in 1965 as An Evening with Batman and Robin, and proved very popular (some theatres showed the chapters as a Saturday matinee). Its success inspired the action-comedy lampoon series Batman (and its 1966 theatrical feature film spin-off) starring Adam West and Burt Ward. 
 **Chunk 5**: * Burt Ward as Dick Grayson / Robin: Batman's sidekick, dubbed the "Boy Wonder": a high school student noted for his recurring interjections in the form of "Holy _, Batman!" 
 LLM response: Burt Ward |

Table 15: Case study on uncertain questions in 2WikiQA.

| Question Id: 0f1ac1a00bdb11eba7f7acde48001122 | |
|---|---|
| Q: **What is the award that the performer of song Sunday Papers earned?** A: Grammy | |
| w/o DENSE | **Chunk 1**: Sunday Papers" is a song written and performed by British new wave musician Joe Jackson. It was released on his debut album, "Look Sharp!".Written as a critique of the British press, "Sunday Papers" features mocking lyrics and reggae inspired music... 
 **Chunk 2**: Caspar Babypants is the stage name of children's music artist Chris Ballew, who is also widely known as the singer of The Presidents of the United States of America. 
 **Chunk 3**: Dáithí Sproule( born 23 May 1950) is a guitarist and singer of traditional Irish music.His niece is the singer Claire Sproule. 
 **Chunk 4**: David Ian "Joe" Jackson (born 11 August 1954) is an English musician and singer-songwriter. Having spent years studying music and playing clubs, Jackson scored a hit with his first release, "Is She Really Going Out with Him?", in 1979. 
 **Chunk 5**: "O Valencia!" is the fifth single by the indie rock bandThe Decemberists, and the first released from their fourth studio album," The Crane Wife".The music was written by The Decemberists and the lyrics by Colin Meloy. 
 LLM response: The award that the performer of song "Sunday Papers" earned is none. |
| DENSE-RAG | **Chunk 1**: Written as a critique of the British press, "Sunday Papers" features mocking lyrics and reggae inspired music. "Sunday Papers" was released as a single in the UK as the follow-up to his single... 
 **Chunk 2**: "Sunday Papers" is a song written and performed by British new wave musician Joe Jackson. It was released on his debut album, "Look Sharp!" 
 **Chunk 3**: Caspar Babypants is the stage name of children's music artist Chris Ballew, who is also widely known as the singer of The Presidents of the United States of America." 
 **Chunk 4**: Dáithí Sproule (born 23 May 1950) is a guitarist and singer of traditional Irish music.His niece is the singer Claire Sproule. 
 **Chunk 5**: He is associated with the 1980s Second British Invasion of the US.He has also composed classical music. He has recorded 19 studio albums and received 5 Grammy Award nominations. 
 LLM response: Joe Jackson earned a Grammy Award nomination. |

