# OpenReview forum: "DENSE-RAG: Measuring and Improving Context Understanding for Consistent Retrieval-Augmented Generation"
_ICLR.cc/2026/Conference — Submitted to ICLR 2026_

### Official Review · Reviewer_hLFs · 2025-10-29

**Soundness:** 1
**Presentation:** 3
**Contribution:** 2
**Rating:** 2
**Confidence:** 3

**Summary:**

The authors propose DENSE, a training-free and model-agnostic method that uses semantically equivalent contexts and degree-based entropy to quantify semantic uncertainty. They further extend it to DENSE-RAG, which enhances RAG through adaptive chunking and context refinement using the DSE metrics, achieving competitive performance without additional training.

**Strengths:**

- The paper proposes a practical metric to measure LLM's understanding of a given context, which can be directly applied to downstream RAG applications
- Extensive experiments and ablation studies support the effectiveness of the proposed metrics

**Weaknesses:**

- The foundation of the work is weak. What is the formal definition of the semantic space $\mathcal{S}$ and the mapping function $\pi$? The paper does not explain how $\pi$ is calculated/approximated. The work relies on the assumption $\pi(q, C) = \pi(q, C')$, but this assumption is not clearly defined. In Section 4.1, the authors claim $\pi(c') = \pi(c)$, where $c'$ is generated by the LLM, but it is unclear how this equality is ensured in practice. This claim also does not align with the statement of Property 1 since we don't know if $\pi(q, c) = \pi(q, c')$. Moreover, the authors aim to evaluate the LLM’s understanding of the given context, yet they simultaneously rely on the LLM’s understanding to generate the rephrasing.
- Is it possible to completely rely purely on the retrieved context when generating the responses during DSE calculation without any parametric knowledge? Since it is not feasible to separate parametric and retrieved knowledge, the response might be grounded to the parametric knowledge rather than the retrieved knowledge.

**Questions:**

Address the above questions.

---

> ### Author Response · Authors · 2025-11-24
>
> W1:
> We appreciate the reviewer’s concern and clarify that S and $\pi$ are not objects we compute explicitly. They are standard abstract formulations used widely in semantics and NLG research (e.g., Kuhn et al., 2023; Farquhar et al., 2024) to express the well-accepted linguistic fact that multiple textual forms may express the same underlying meaning. Our intent is descriptive rather than definitional—similar to how semantic entropy literature uses semantic clusters without defining an explicit metric space.
> Formally, we do not require computing $\pi$ or operating in an explicit semantic space. Instead, we operationalize the assumption $\pi(c)=\pi(c')$ using empirical verification:
>
> In Section 6, we show that LLM responses under rephrased chunks exhibit no significant QA performance drift, demonstrating that rephrasing preserves answer-relevant semantics.
> We additionally evaluate rephrasing using DeBERTa-large MNLI and the LLM itself for bidirectional entailment, showing that 95% of rephrased chunks are semantically equivalent to the originals (Table 6). This provides empirical grounding for $\pi(c)=\pi(c')$ without requiring explicit computation of $\pi$.
>
> |Method|TriviaQA|NQ|AmbigNQ|2WikiQA|
> |-|-|-|-|-|
> |LLM-as-a-judge|98.37%|95.95%|97.88%|93.24%|
> |Deberta-as-a-judge|97.65%|98.28%|96.35%|98.40%|
>
> Regarding Property 1, our claim is simply the standard principle in consistency-based LLM evaluation: **If two contexts are semantically equivalent under question q, a model that understands the context should produce semantically equivalent answers.**
> This is precisely the assumption used in semantic consistency, semantic entropy, and numerous QA consistency works. We follow the same convention.
> Finally, there is no circularity in using the LLM for rephrasing while evaluating its contextual understanding:
>
> 1. Rephrasing is strictly easier than QA—it only requires syntactic transformation.
>
> 2. Rephrasing does not require “deep semantic reasoning”; QA does.
>
> 3. We empirically validate that rephrasing preserves semantics (entailment test + QA performance test).
>
> Thus, rephrasing using the same LLM does not undermine the goal of measuring understanding; instead, it ensures controlled perturbations while preserving semantics.
>
> W2:
> We respectfully clarify that no RAG system, including all prior work (RePlug, RankRAG, Self-RAG, etc.) operates **“purely without parametric knowledge”**. RAG by definition combines parametric knowledge inside a pre-trained LLM and retrieved evidence. Our goal is not to remove parametric knowledge (which is impossible), but to control for its randomness and measured variation reflects differences in the retrieved context.
> To address this, we use greedy decoding to eliminate sampling noise so that: 1. If the retrieved context is unchanged, the LLM always produces an identical answer. 2. Any change in the output necessarily arises from the changes in the context, not from parametric knowledge.
>
> Thus, our method does not attempt to separate parametric and retrieved knowledge (which is theoretically impossible), but instead fixes the parametric component and measures the sensitivity of outputs to controlled semantic perturbations in the retrieved context.
>
> In short:
> 1. RAG inherently uses both parametric and retrieval knowledge.
>
> 2. We do not require eliminating parametric knowledge; we only require eliminating its variability.
>
> 3. Under this controlled setting, DENSE isolates the contribution of retrieved context formatting/semantics to output stability.

---

### Official Review · Reviewer_EmPP · 2025-11-01

**Soundness:** 2
**Presentation:** 3
**Contribution:** 3
**Rating:** 6
**Confidence:** 2

**Summary:**

This paper asks a simple question: “Does the model actually understand the stuff we retrieved?” The authors create several rephrased versions of each retrieved chunk, run the LLM on these semantically equivalent contexts with greedy decoding, and then measure how much the answers’ meanings wobble; more wobble ⇒ poorer context understanding. They turn that idea into a metric (DSE) and a plug-in pipeline that first measures uncertainty and then, only when needed, fixes the context via semantic chunking and an iterative prune-and-refill step.

**Strengths:**

1. DSE is a nice graph view of semantic agreement across answers; skipping clustering and using degree over an entailment matrix makes the signal straightforward to compute and attribute back to specific chunks.


2. The paper shows a consistent pattern that higher DSE aligns with worse QA, which gives DSE a real “thermometer” feel for context understanding rather than just another heuristic.

**Weaknesses:**

1. The pipeline uses an LLM to rephrase chunks and another LLM to judge entailment between answers; this makes the measurement depend on two models’ quirks, so I’d like to see robustness when swapping the NLI judge or using a lightweight supervised NLI model.

2. “Semantically equivalent” is enforced by a prompt rather than a guarantee; please show automatic checks (e.g., NLI on chunk pairs) that the rephraser didn’t drop constraints, especially on math, dates, and entity coreference.

3. Greedy decoding is tidy for measurement, but production RAG often samples; if decoding changes, does DSE still track understanding, or does sampling noise blur the signal.

4. Single-chunk edits help attribution but miss cross-chunk interactions (e.g., a coreference in chunk A that only makes sense with chunk B), which could inflate DSE for the wrong reason.

**Questions:**

1. How do you verify that chunk rephrasings preserve constraints (numbers, negations, entity links) beyond qualitative checks; any automated guardrails before a rephrased chunk enters DENSE.

2. If we replace the LLM-as-NLI with a small supervised NLI (e.g., DeBERTa) or a cosine-semantic scorer, does DSE rank examples similarly and still predict failure; please report rank correlation.

---

> ### Author Response · Authors · 2025-11-24
>
> W1&Q2:
> To evaluate the sensitivity to the NLI backbone, we additionally replaced the LLM-based judge with an independent supervised NLI model DeBERTa-large-mnli. We add experimental results in section 6.2.1. These results demonstrate that DENSE does not rely on self-reinforcing agreement from a single LLM.
>
> W2&Q1:
> First we have validated that Rephrased chunks is semantically equivalent by comparing RAG performance with and without DENSE. Additionally, for each rephrased chunks, we performed semantic-equivalence verification between the original chunks and rephrased ones. We use both LLM and DeBERTa to check the semantic entailment. Above 95% rephrased chunks are identified as semantic entailment by both LLM and DeBERTa. We add experiment results and discussion in section 4.1 and table 6 in Appendix D.
> |Method|TriviaQA|NQ|AmbigNQ|2WikiQA|
> |-|-|-|-|-|
> |LLM-as-a-judge|98.37%|95.95%|97.88%|93.24%|
> |Deberta-as-a-judge|97.65%|98.28%|96.35%|98.40%|
>
> W3:
> Our use of greedy decoding was deliberate: sampling introduces randomness from the LLM itself, which can confound the effect we aim to measure, semantic variation caused by perturbations in the retrieved context. Greedy decoding therefore ensures that DENSE reflects context-understanding rather than temperature-induced randomness. To assess robustness under realistic sampling conditions, we conduct additional experiments with temperature ∈ {0.25, 0.5, 1, 3, 5,7 }. And we implement 5 semantic based uncertainty methods on four dataset (new added table 1 and table 8). We demonstrate that DENSE continues to be an effective uncertainty indicator even when sampling is used in practice. We show some results here, and full results are added in section 6.2 and Appendix I.1.
>
> |Method|TriviaQA||NQ||AmbigNQ||2WikiQA||
> |-|-|-|-|-|-|-|-|-|
> ||AUROC|AUARC|AUROC|AUARC|AUROC|AUARC|AUROC|AUARC|
> |greedy encoding||||||||
> |DENSE_llm|75.49|92.69|67.26|75.83|68.20|78.06|65.02|66.27|
> |DENSE_deberta|66.94|91.02|59.42|73.69|58.01|74.89|53.49|61.42|
> |temperature=0.25|||||
> |Discrete Semantic Entropy|64.07|88.72|61.33|72.29|60.40|75.94|57.28|60.80|
> |U_eigv|66.64|89.05|64.16|73.08|62.90|76.01|57.37|60.43|
> |U_deg|66.56|89.03|64.05|73.69|62.74|76.79|57.25|60.82|
> |KLE heat_t=0.1|66.40|89.00|63.93|73.89|62.53|75.84|56.99|60.36|
> |KLE deberta matern_\kappa=3.0,\nu=3.0|63.97|89.11|61.98|72.51|62.33|77.56|56.78|61.05|
> |DENSE_llm(ours)|**75.63**|**92.52**|**67.96**|**75.10**|**69.94**|**79.54**|**65.74**|**66.38**|
> |temperature=0.5|
> |Discrete Semantic Entropy|71.15|90.56|65.59|74.46|66.61|78.79|61.50|63.57|
> |U_eigv|74.32|91.90|**69.35**|**77.08**|68.79|77.47|61.97|63.93|
> |U_deg|74.24|91.53|69.28|76.82|68.64|77..89|61.84|63.44|
> |KLE heat_t=0.1|73.87|91.48|69.18|76.79|68.04|77.78|61.41|63.72|
> |KLE deberta matern_\kappa=3.0,\nu=3.0|68.84|90.74|64.57|75.09|63.25|77.95|60.17|63.54|
> |DENSE_llm(ours)|**77.53**|**93.06**|69.23|76.71|**72.02**|**81.28**|**67.28**|**68.08**|
>
>
> W4:
> We thank the reviewer for raising this insightful point. Our design intentionally rephrases one chunk at a time to isolate the model’s semantic sensitivity to that specific chunk. This single-chunk perturbation is essential for attributing uncertainty to a specific piece of retrieved evidence—otherwise, interactions across multiple edited chunks would introduce confounding factors and make attribution ambiguous. We completely agree that cross-chunk interactions (e.g., coreference or multi-sentence semantics spanning different chunks) may also influence the model’s understanding. Extending DENSE to detect cross-chunk dependencies could offer an interesting view of context understanding. We will include this as an important direction for future work and discuss the potential impact of cross-chunk semantic interaction in the revised version.

---

### Official Review · Reviewer_esQU · 2025-11-01

**Soundness:** 1
**Presentation:** 1
**Contribution:** 2
**Rating:** 2
**Confidence:** 4

**Summary:**

The paper proposes a method called DENSE to assess whether a given LLM "understand" the retrieved set of chunks, i.e., context, and risks generating an incorrect answer. The method consists in prompting the target LLM several times with several paraphrases (rephrased sub-chunks) and gather the set of generated answers; if there is variation in the collected answer set then there is uncertainty otherwise not (the authors define a DENSE score and a threshold for it).  In addition, the paper proposes two ways to use DENSE to improve retrieval augmented QA. One is incremental chunk extension that iteratively adds sentences to initially retrieved chunks. The other one is using DENSE to analyse chunks in the set as certain or uncertain, and this last as necessary/unnecessary. Then the set of chunks is refined by removing and adding new chunks as necessary.

**Strengths:**

- Determining uncertainty via chunk paraphrasing and answer variation is an interesting alternative (though not completely new).

**Weaknesses:**

- The proposed approach seems to be rather expensive, both the context understanding (uncertainty detection) and QA performance improvement methods. Several passes over the chunks are required. It would be useful if the authors could add a discussion and report the number of calls the approach requires.

- The DENSE approach for context understanding evaluation should be compared to uncertainty estimation approaches (e.g., Perplexity, [Kuhn et al., 2023] see for a comprehensive library [1]).

[1] https://arxiv.org/abs/2406.15627

- The incremental sentence  extension of chunks seems to work well but is not clear how the documents used are obtained.

- The comparisons with existing methods in Table 2 seem weak. AmbigQA has only one reported value. Also, what retrieval source are these previous work using? Unfortunately, different paper often use different corpora (e.g.,  gold+distractors of the original dataset, different Wikipedia dumps, different retrievers). So it would be useful to have this clarified for each of the reported systems. Also, it would be useful to have a discussion of the chosen comparison method and how it compares to the proposes in the paper.

**Questions:**

- Line 021, what is "degree-based" entropy? Line 221, what is the "degree" of a response. It is not clear what the authors mean by degree.

- Section 4.2, what is the difference between degree-based uncertainty and [Kuhn et al., 2023]? The explanation at paragraph starting at Line 203 is not clear. Perhaps an example would help.

- How many paraphrases are generated for each chunk to determine an answer set? Which model is used for paraphrasing?

- Adaptive semantic chunking (Section 5.1) is not clearly explained. At Line 249, "we merge all documents", which documents are these? From what does the iterative aggregation of sentences starts? Each initially retrieved chunk is augmented with sentences iteratively?

- The context refinement step is similar to re-ranking, maybe it should be compared with re-ranking methods specifically, e.g., baseline ones like retrieving a large number of chunks and the using NLI.

---

> ### Author Response · Authors · 2025-11-24
>
> W1: DENSE-RAG is designed for a zero-training, model-agnostic, black-box deployment scenario, where users only have access to an LLM API and cannot fine-tune a retriever, train a reranker, or host additional models. In this regime, the dominant cost is the per-query LLM call budget.
> **We have already included this analysis in the conclusion and appendix L. To make the discussion easier to locate for reviewers, we have also moved the part concerning complexity to Section 6.3.**But we will clarify the fact that the multi-pass refinement is selective: only chunks flagged as high-uncertainty by DENSE are expanded, so the average cost per query is substantially lower than the worst-case.
>
> W2: We appreciate the reviewer’s suggestion. Approaches such as semantic entropy(Kuhn et al., 2023) are designed to quantify LLM’s intrinsic generation uncertainty in natural-language generation, not RAG task. In contrast, **DENSE targets a fundamentally different objective: it evaluates LLM’s context-understanding by measuring how the LLM’s answer changes when the retrieved context is perturbed while semantics fixed.** Moreover, DENSE provides chunk-level attribution, enabling us to locate which retrieved segment contributes to the instability, which semantic entropy like methods are not designed to do. Because the two classes of methods operate in different regimes, a direct comparison is not theoretically aligned. But we agree that reporting baseline of semantic uncertainty scores is valuable for illustrating the distinction and better to show our methods effectiveness. We implement 5 semantic based uncertainty methods on four dataset (new added table 1 and table 8). We demonstrate that DENSE continues to be an effective uncertainty indicator even when sampling is used in practice. We show part of experimental result here, full results are added in section 6.2 and Appendix I.1.
>
> |Method|TriviaQA||NQ||AmbigNQ||2WikiQA||
> |-|-|-|-|-|-|-|-|-|
> ||AUROC|AUARC|AUROC|AUARC|AUROC|AUARC|AUROC|AUARC|
> |greedy encoding||||||||
> |DENSE_llm|75.49|92.69|67.26|75.83|68.20|78.06|65.02|66.27|
> |DENSE_deberta|66.94|91.02|59.42|73.69|58.01|74.89|53.49|61.42|
> |temperature=0.25|||||
> |Discrete Semantic Entropy|64.07|88.72|61.33|72.29|60.40|75.94|57.28|60.80|
> |U_eigv|66.64|89.05|64.16|73.08|62.90|76.01|57.37|60.43|
> |U_deg|66.56|89.03|64.05|73.69|62.74|76.79|57.25|60.82|
> |KLE heat_t=0.1|66.40|89.00|63.93|73.89|62.53|75.84|56.99|60.36|
> |KLE deberta matern_\kappa=3.0,\nu=3.0|63.97|89.11|61.98|72.51|62.33|77.56|56.78|61.05|
> |DENSE_llm(ours)|**75.63**|**92.52**|**67.96**|**75.10**|**69.94**|**79.54**|**65.74**|**66.38**|
> |temperature=0.5|
> |Discrete Semantic Entropy|71.15|90.56|65.59|74.46|66.61|78.79|61.50|63.57|
> |U_eigv|74.32|91.90|**69.35**|**77.08**|68.79|77.47|61.97|63.93|
> |U_deg|74.24|91.53|69.28|76.82|68.64|77..89|61.84|63.44|
> |KLE heat_t=0.1|73.87|91.48|69.18|76.79|68.04|77.78|61.41|63.72|
> |KLE deberta matern_\kappa=3.0,\nu=3.0|68.84|90.74|64.57|75.09|63.25|77.95|60.17|63.54|
> |DENSE_llm(ours)|**77.53**|**93.06**|69.23|76.71|**72.02**|**81.28**|**67.28**|**68.08**|
>
>
> W3&Q4:
> **No additional documents are introduced during adaptive semantic chunking. No external corpus or auxiliary documents are used.** When DENSE is high during recursive chunks, we will run semantic chunking. The adaptive chunking procedure is:
> 1.	Sentence-level segmentation. For each retrieved document, we split it into a sequence of sentences.
> 2.	Iterative aggregation within the same document. Starting from the first sentence s_1, we compute its embedding similarity with the next sentence s_2. If the similarity exceeds a threshold and the resulting chunk does not exceed the maximum token limit, we merge them. Otherwise, s_2 begins a new chunk. This process continues sequentially until the end of the document, yielding a set of semantic chunks \{c_1, c_2, c_n\}. Aggregation happens only within the boundaries of each retrieved document; we do not mix text across documents.
>
> Thus, adaptive chunking changes only the segmentation strategy, not the corpus. We will clarify this explicitly in the revision.

---

> > ### Author Response · Authors · 2025-11-24
> >
> > W4: First, we would like to emphasize that AmbigQA, as an ambiguous question set, plays a unique and irreplaceable role in evaluating RAG uncertainty and measuring context understanding. This has also been highlighted in multiple survey papers, such as [1, 2, 3].
> > Moreover, our primary emphasis is on the **vertical comparison brought by DENSE-RAG: how the proposed module enhances LLM performance on uncertain questions**. The horizontal comparison with other RAG baselines is provided merely as contextual reference, not as the core contribution of this work.
> > We thank the reviewer for raising this challenging but common issue in today’s AI research. Unfortunately, existing systems differ widely in their retrieval pipelines: different retrievers, different Wikipedia dumps, or fine-tuned LLMs, making direct uniform comparison is difficult. To ensure fairness and reproducibility, we follow three strict principles:
> > 1. **We use only the dataset-provided official corpora**(e.g., the official corpora that ships with TriviaQA, NQ). No external documents or alternative knowledge sources are introduced.
> > 2. **We avoid any retriever fine-tuning**. All our experiments use a lightweight, off-the-shelf encoder (UAE) without any task-specific adaptation, so our improvements do not come from retriever optimization.
> > 3. **We strictly follow previous work in data split, perform no query preference**. For TriviaQA and NaturalQuestion, we follow the widely used KILT benchmark split. And for AmbigNQ and 2WikiQA, we use the dev set. We never operate like “sample first 500 questions”.
> > Because most prior baselines do not provide a unified retriever version that can be re-run on a common corpus, we try our best to distinguish them by report which baselines use (i) fine-tuned retrievers, (ii) fine-tuned generators in table 2. Our baseline setup (retriever, generator, corpus, data splits) is detailed in Appendix H and I, and we will expand this section in the revision to clarify the retrieval details.
> >
> >
> > Q1:
> > In DENSE, we treat the set of generated responses under semantically equivalent contexts as a graph: each response is a node, and an edge is added between two nodes if their semantic relationship (via NLI) is entailment in both directions. The degree of a response represents how many other responses are semantically aligned with it in the graph, corresponding to the number of adjacent nodes in the semantic graph.
> >
> > Degree-based entropy summarizes this structure by measuring how concentrated or dispersed the semantic consistency is across responses. A high-degree node indicates stable, semantically consistent outputs; low degree indicates semantic divergence. We will revise the paper to explicitly introduce this graph representation and define degree and degree-based entropy more clearly.
> >
> > Q2:
> > We will revise this section and add an illustrative example in figure 1. Conceptually, degree-based uncertainty and semantic-entropy methods differ in two essential ways:
> >
> > 1.	**Source of uncertainty**: Kuhn et. al measures the model’s intrinsic generation uncertainty, typically estimated through sampling or beam-level variation under the same context. In contrast, DENSE measures context-understanding uncertainty by observing how the model’s answer changes under semantically equivalent perturbations of the retrieved context, computed under greedy decoding to remove parametric-generation noise.
> >
> > 2.	**Granularity and attribution**: Semantic-uncertainty methods assign one uncertainty score per query, but cannot distinguish which part of the retrieval context caused answer instability. DENSE instead constructs a graph over responses generated from paraphrases of individual chunks, yielding chunk-level uncertainty scores ("degree" and "degree-based entropy"). This allows us to identify which retrieved segments are responsible for semantic shifts in the answer.
> >
> > Q3:
> > For each chunk, we generate one paraphrased version for each chunk. Given k retrieved chunks, this yields k paraphrased contexts. Paraphrasing is performed using the same LLM as the inference model, to ensure semantic unchanged while keeping the pipeline single-model and zero-training.
> > To guarantee that paraphrasing does not introduce semantic drift, we validate semantic judges: (i) the LLM-based NLI prompt and (ii) DeBERTa-large-MNLI. Across datasets, we observe that 95% of pairs satisfy bi-directional entailment. Full results are shown in Appendix D.

---

> ### Author Response · Authors · 2025-11-24
>
> Q5:
> We thank the reviewer for pointing out the connection between our context refinement step and “re-ranking”. In the current landscape of RAG research, it has become increasingly difficult to isolate the effect of reranking from other post-retrieval operations. Many recent approaches enhance, replace, or refine retrieved documents using auxiliary models, LLM fine-tuning, or in-context learning strategies to improve QA performance. As highlighted in the most highly cited RAG survey [4], post-retrieval processing is now a standard component across both “advanced RAG’’ and “modular RAG’’ paradigms. For this reason, we choose to explicitly annotate in tables whether each compared method uses a retriever or generator that has been fine-tuned specifically for the QA task, allowing readers to better understand the methodological differences and sources of performance gains.
>
> [1]A survey on uncertainty quantification of large language models: Taxonomy, open research challenges, and future directions
>
> [2]A survey on rag meeting llms: Towards retrieval-augmented large language models.
>
> [3]Uncertainty quantification and confidence calibration in large language models: A survey
>
> [4]Retrieval-Augmented Generation for Large Language Models: A Survey

---

### Official Review · Reviewer_XtSK · 2025-11-02

**Soundness:** 2
**Presentation:** 2
**Contribution:** 2
**Rating:** 4
**Confidence:** 4

**Summary:**

The paper introduces DENSE (Degree-based Semantic Entropy), a training-free, model-agnostic estimator of contextual understanding in RAG systems. DENSE measures semantic uncertainty by generating responses under semantically equivalent (rephrased) contexts and computing a graph-based degree score over pairwise NLI entailments between responses. Building on this signal, the paper proposes DENSE-RAG, which (i) selectively switches from fixed-size to semantic chunking for uncertain questions (Adaptive Semantic Chunking), and (ii) performs Iterative Context Refinement by classifying retrieved chunks into certain/necessary/unnecessary and pruning/refilling accordingly. Experiments on TriviaQA, NQ, AmbigNQ, and 2WikiQA (and a preliminary CNN/DM study) show strong correlation between DENSE and accuracy and competitive or superior performance to several SOTA methods without any finetuning.

**Strengths:**

1. DENSE-RAG introduces a lightweight, plug-and-play uncertainty estimation framework that requires no additional training or fine-tuning, making it practical for real-world black-box LLMs.
2. Strong evidence that higher DENSE values correlate with reduced QA accuracy across datasets and model sizes, validating it as a reliable proxy for contextual understanding.
3. Includes detailed ablations, sensitivity tests, runtime analysis, and clear pseudocode, improving reproducibility and interpretability.

**Weaknesses:**

1. The same LLM is used for rephrasing, inference, and entailment, which risks internal bias and inflates semantic agreement.
2. The paper reports runtime complexity but lacks cost comparisons against strong fine-tuned baselines (e.g., RankRAG, RAG-DDR), leaving efficiency claims incomplete.
3. Experiments focus mostly on QA tasks with preliminary summarization results; effectiveness on other domains or structured reasoning tasks remains unclear.

**Questions:**

1. Which NLI model and thresholds were used for entailment evaluation, and how sensitive are the results to changing the NLI backbone?
2. How does DENSE perform under non-greedy decoding settings (e.g., temperature sampling) where response variance is higher?
3. Can DENSE-RAG be adapted for multi-turn or conversational RAG settings, and how would uncertainty accumulation across turns be handled?

---

> ### Author Response · Authors · 2025-11-24
>
> W1: To address reviewers concern that using the same LLM for paraphrasing and entailment may introduce self-agreement, potentially inflating semantic consistency, we conducted two verification experiments:
>
> 1.	For each rephrased chunks, we performed semantic-equivalence verification between the original chunks and rephrased ones. We use both LLM and DeBERTa to check the semantic entailment. Above 95% rephrased chunks are identified as semantic entailment by both LLM and DeBERTa. We add experiment results and discussion in section 4.1 and table 6 in Appendix D.
> |Method|TriviaQA|NQ|AmbigNQ|2WikiQA|
> |-|-|-|-|-|
> |LLM-as-a-judge|98.37%|95.95%|97.88%|93.24%|
> |Deberta-as-a-judge|97.65%|98.28%|96.35%|98.40%|
> 2.	We change the NLI tool to a supervised pretrain NLI model DeBERTa-large-mnli, to replace LLM in DENSE, DENSE remains robust under NLI methods changes. We add experimental results in section 6.2.1
> These results demonstrate that DENSE does not rely on self-reinforcing agreement from a single LLM.
>
>
> W2:
> We appreciate the reviewer’s suggestion. However, fine-tuned RAG pipelines (e.g., RankRAG, RAG-DDR) operate in a different cost regime: they require (i) training a reranker or retriever, or (ii) fine-tuning LLM on QA data. In contrast, **DENSE-RAG is adapted for strict black-box settings, where only a single LLM API is accessible and no additional training or models are allowed.**
>
> Because the two regimes optimize under fundamentally different constraints (trainable vs. non-trainable), their absolute runtime cost is not directly comparable. We therefore report per-query LLM call complexity in Appendix L. We clarify this distinction in the revised text and add a discussion contrasting “train-heavy” vs “zero-training” pipelines. Our goal is not to compare with fine-tuned systems in runtime complexity, but to provide a robust model-agnostic uncertainty measurement that works even in a black-box LLM environment.
>
> W3&Q3:
> We thank the reviewer for highlighting the question of generality. Our experiments focus primarily on open-book QA. However, DENSE itself is not task-specific: by construction, it measures how semantically stable the model’s output remains under semantically equivalent perturbations of the retrieved evidence. This principle applies to any retrieval-augmented generation task where answers needed be grounded in context.
>
> Q1:
> Entailment was performed using the LLM itself with a structured prompt: given the original chunk c and its paraphrase c′, the LLM was asked to classify the relation as *entailment / contradiction / neutral*. To evaluate the sensitivity to the NLI backbone, we additionally replaced the LLM-based judge with an independent supervised NLI model DeBERTa-large-mnli. DENSE maintains consistently strong performance across datasets under Deberta NLI backbone. The experiment results is added in section 6.2.1.
>
> Q2:
> Our use of greedy decoding was deliberate: sampling introduces randomness from the LLM itself, which can confound the effect we aim to measure, semantic variation caused by perturbations in the retrieved context. Greedy decoding therefore ensures that DENSE reflects context-understanding rather than temperature-induced randomness.
>
> To assess robustness under realistic sampling conditions, we conduct additional experiments with temperature ∈ {0.25, 0.5, 1, 3, 5,7 }. And we implement 5 semantic based uncertainty methods on four dataset (new added table 1 and table 8). We demonstrate that DENSE continues to be an effective uncertainty indicator even when sampling is used in practice. Here we show the result under 0.25 and 0.5 temprature, full results are added in section 6.2 and Appendix I.1
>
> |Method|TriviaQA||NQ||AmbigNQ||2WikiQA||
> |-|-|-|-|-|-|-|-|-|
> ||AUROC|AUARC|AUROC|AUARC|AUROC|AUARC|AUROC|AUARC|
> |greedy encoding||||||||
> |DENSE_llm|75.49|92.69|67.26|75.83|68.20|78.06|65.02|66.27|
> |DENSE_deberta|66.94|91.02|59.42|73.69|58.01|74.89|53.49|61.42|
> |temperature=0.25|||||
> |Discrete Semantic Entropy|64.07|88.72|61.33|72.29|60.40|75.94|57.28|60.80|
> |U_eigv|66.64|89.05|64.16|73.08|62.90|76.01|57.37|60.43|
> |U_deg|66.56|89.03|64.05|73.69|62.74|76.79|57.25|60.82|
> |KLE heat_t=0.1|66.40|89.00|63.93|73.89|62.53|75.84|56.99|60.36|
> |KLE deberta matern_\kappa=3.0,\nu=3.0|63.97|89.11|61.98|72.51|62.33|77.56|56.78|61.05|
> |DENSE_llm(ours)|**75.63**|**92.52**|**67.96**|**75.10**|**69.94**|**79.54**|**65.74**|**66.38**|
> |temperature=0.5|
> |Discrete Semantic Entropy|71.15|90.56|65.59|74.46|66.61|78.79|61.50|63.57|
> |U_eigv|74.32|91.90|**69.35**|**77.08**|68.79|77.47|61.97|63.93|
> |U_deg|74.24|91.53|69.28|76.82|68.64|77..89|61.84|63.44|
> |KLE heat_t=0.1|73.87|91.48|69.18|76.79|68.04|77.78|61.41|63.72|
> |KLE deberta matern_\kappa=3.0,\nu=3.0|68.84|90.74|64.57|75.09|63.25|77.95|60.17|63.54|
> |DENSE_llm(ours)|**77.53**|**93.06**|69.23|76.71|**72.02**|**81.28**|**67.28**|**68.08**|

---

### Meta-Review · Area_Chair_4Vmp · 2025-12-23

**Summary:**

### Summary
The paper proposes DENSE, a training-free, model-agnostic signal for context understanding in RAG by paraphrasing retrieved chunks into semantically equivalent variants, measuring semantic wobble of answers, and computing a degree-based semantic entropy from an entailment graph over responses.

### Reviewer Summary

Strengths: the question is well-motivated, the metric is plug-and-play, and experiments show DENSE correlates with QA failures while DENSE-RAG gives competitive gains in a strict training-free setting

Key concerns: costbudget may be high; using the same LLM for inference can create self-agreement bias; positioning vs prior uncertainty and reranking methods is still not fully solid; theory is largely “assumption + empirical validation”; coverage is QA-centric and cross-chunk interactions remain unhandled.

### AC comments

The rebuttal meaningfully addresses several core concerns—showing robustness to an independent NLI judge (DeBERTa MNLI), verifying paraphrase equivalence, and demonstrating the uncertainty signal under sampling settings.
However, two issues remain decisive: (i) cost + comparability—even if the paper targets a black-box/zero-training regime, the practical cost and per-query call budget are still hard to judge and not easily comparable; (ii) positioning and rigor—the semantic-equivalence premise remains largely empirical, cross-chunk dependencies are acknowledged but not handled, and generality beyond QA is still mostly aspirational. Given the mixed reviews with two strong rejects, I do not find the current evidence and framing stable enough for acceptance.

**Reviewer Concerns:**

### Reviewer XtSK

* Resolved: self-agreement risk from using the same LLM; robustness beyond greedy decoding.
* Unresolved: broader generality beyond QA and the overall “cost regime” remains unclear.

### Reviewer esQU

* Resolved: missing comparisons to uncertainty baselines; unclear “degree-based” definitions; unclear adaptive chunking mechanics.
* Unresolved: pipeline still looks expensive; apples-to-apples comparisons remain hard due to different corpora/retrievers; refinement vs reranking remains largely narrative.

### Reviewer EmPP

* Resolved: robustness to NLI judge swap; paraphrase constraint preservation; sampling robustness.
* Unresolved: cross-chunk interactions are not captured by single-chunk perturbation.

### Reviewer hLFs

* Unresolved: justification that “semantic space” is abstract and need not be computed explicitly.
* Unresolved: perceived circularity (LLM paraphrases and is evaluated); impossibility of separating parametric vs retrieved knowledge.

**Reviewer Scores:**

* XtSK (4): likely unchanged.
* EmPP (6): likely unchanged.
* esQU (2): may rise to 4, but cost/comparability concerns likely remain.
* hLFs (2): likely unchanged given the focus on rigor/circularity.

---

### Decision · Program_Chairs · 2026-01-26

Reject